# Provably Improving Generalization of Few-Shot Models with Synthetic Data

**Lan-Cuong Nguyen** [1 2]  **Quan Nguyen-Tri** [1]  **Bang Tran Khanh** [3]  **Dung D. Le** [3]  **Long Tran-Thanh** [4]  **Khoat Than** [2]

## Abstract

Few-shot image classification remains challenging due to the scarcity of labeled training examples. Augmenting them with synthetic data has emerged as a promising way to alleviate this issue, but models trained on synthetic samples often face performance degradation due to the inherent gap between real and synthetic distributions. To address this limitation, we develop a theoretical framework that quantifies the impact of such distribution discrepancies on supervised learning, specifically in the context of image classification. More importantly, *our framework suggests practical ways to generate good synthetic samples and to train a predictor with high generalization ability*. Building upon this framework, we propose a novel theoretical-based algorithm that integrates prototype learning to optimize both data partitioning and model training, effectively bridging the gap between real few-shot data and synthetic data. Extensive experiments results show that our approach demonstrates superior performance compared to state-of-the-art methods, outperforming them across multiple datasets.

## 1. Introduction

Deep learning models often require extensive annotated datasets to achieve excellent performance. However, creating such datasets is both labor-intensive and costly. To address this challenge, many recent studies has explored synthetic data as an alternative approach to training deep learning models.

In this paper, we focus on a specific scenario: *How to efficiently use a synthetic dataset along with few real samples to train a predictor in a downstream task?* A naive way of using synthetic data can degrade performance on downstream tasks (Van Breugel et al., 2023). The main reason comes from *the distribution gap*, which is the discrepancy between the real and synthetic data distributions, due to the imperfect nature of the generator for a downstream task. Overcoming this gap is crucial for ensuring effective utilization of synthetic data in this training scenario.

Some recent methods minimize this gap by fine-tuning generative models. Notable examples include RealFake (Yuan et al., 2024) and DataDream (Kim et al., 2024) which focus on full dataset and few-shot image classification, respectively. These approaches emphasize pixel-space distribution matching. The importance of feature-space alignment has been demonstrated in dataset distillation (Wang et al., 2018), where condensed datasets are created to approximate the performance of models trained on full datasets (Wang et al., 2022; Zhao & Bilen, 2023; Zhao et al., 2023). An issue of distribution matching is the potential for incorrect associations between samples from different classes. DataDream (Kim et al., 2024) partially mitigates this by fine-tuning generators for each class independently, yet its results remain suboptimal. A promising direction to address this issue is prototype learning, which emphasizes local class-specific behavior. Prototype-based approaches have been shown to enhance performance in various applications, such as robust classification (Yang et al., 2018), data distillation (Su et al., 2024), and dataset representation (Tu et al., 2023; van Noord, 2023). While such recent methods have shown promise in narrowing the distribution gap, they predominantly rely on heuristics, lacking theoretical guarantees.

In more detail, for a downstream task, we are interested in these questions:

*1. What properties can indicate the goodness of a synthetic dataset?*
*2. How to generate a good synthetic dataset?*
*3. How to efficiently train a predictor from a training set of both real and synthetic samples?*
*4. How can the quality of a generator affect the generalization ability of the trained predictor?*

From the theoretical perspective, existing studies typically have preliminary answers to simple models only. Indeed, Yuan et al. (2024) proposed a framework that generates synthetic samples by training/finetuning a generator to minimize the distribution gap, addressing the second question.

---

[1]FPT Software AI Center [2]Hanoi University of Science and Technology [3]VinUniversity [4]University of Warwick. Correspondence to: Khoat Than <khoattq@soict.hust.edu.vn>.

*Proceedings of the 42^{nd} International Conference on Machine Learning*, Vancouver, Canada. PMLR 267, 2025. Copyright 2025 by the author(s).

Räisä & Honkela (2024) and Zheng et al. (2023) investigated the first question and suggested that synthetic data should be close to real data samples. Recently, Ildiz et al. (2025) provided a systematic study about the role of surrogate models which create the synthetic labels for training *linear models*. However, as they only focus on linear models, their study cannot provide a reasonable answer to the four above-mentioned questions of interest.

Against this background, in this paper we make a systematic step towards understanding the role of synthetic data and distribution for training a predictor in a few-shot setting. Specifically, our contributions are as follows:

- *Theory:* We analyze the generalization ability of a predictor, trained with both synthetic and real data. A novel bound on the test error of the predictor is presented. It suggests that a good synthetic set not only should be *close to the real samples*, but also should be *diverse* so that the trained predictor can be locally robust around the training samples. This bound also suggests a theoretical principle to generate good synthetic samples, and a practical way to train a predictor that generalizes well on unseen data.

- We further present two novel bounds on the test error. Those bounds encloses the discrepancy between the true and synthetic distributions into account. It theoretically shows that the closer the synthetic distribution, the better the trained predictor becomes. Hence our analyses provide theoretical answers to the four questions above.

- *Methodology*: Guided by our theoretical bounds, we introduce a novel loss function and training paradigm designed to jointly optimize data partitioning and model training, effectively minimizing generalization errors.

- *Empirical Validation*: We evaluated our method in the context of few-shot image classification using synthetic data. Experimental results demonstrate that our method consistently outperforms state-of-the-art methods across multiple datasets.

*Organization:* The next section summarizes the related work. In Section 3, we investigate the theoretical benefits of synthetic data and the quality of a synthetic distribution to train a model. Those investigations suggest explicit answers to the four questions mentioned above. Section 4 discusses how to train a good few-shot model, while Section 5 reports our main experimental results. Mathematical proofs and additional experimental results can be found in the appendices.

## 2. Related Works

### 2.1. Generative Data Augmentation

With the rapid advancements in generative models, training with synthetic data has gained significant attention. In the context of image classification, recent studies have employed text-to-image models, aligning conditional distributions through text-prompt engineering. These efforts have explored class-level descriptions (He et al., 2023), instance-level descriptions (Lei et al., 2023), and lexical definitions (Sariyildiz et al., 2022). While text-based conditioning offers flexibility, it often overlooks intrinsic visual details, such as exposure, saturation, and object-scene co-occurrences. To address these shortcomings, RealFake (Yuan et al., 2024) attempts to reduce the distribution misalignment with a model-agnostic method that minimizes the maximum mean discrepancy between real and synthetic distributions by fine-tuning the generator. This discrepancy naturally becomes a lower bound for the training loss of diffusion models under certain conditions. GenDataAgent (Li et al., 2025) further improves diversity by perturbing image captions and quality by filtering out with the Variance of Gradient (VoG) score.

Beyond empirical findings, theoretical frameworks have been also developed to validate the use of synthetic data. Zheng et al. (2023) employed algorithmic stability to derive generalization error bounds based on the total variation distance between real and synthetic image distributions. Additionally, Gan & Liu (2025) analyzed the impact of synthetic data on post-training large language models (LLMs) using information theory. However, none of these results can be adopted to the few-shot learning domain.

### 2.2. Few-shot Image Classification with Vision-Language Models

The rising performance of vision-language models, such as CLIP (Radford et al., 2021), has sparked interest in applying them to few-shot image classification. Initial methods focused on leveraging visual or textual prompting (Jia et al., 2022; Zhou et al., 2022; Khattak et al., 2023; Li et al., 2024; Yao et al., 2024; Zheng et al., 2024). Another simultaneous line of research focused on leveraging parameter-efficient fine-tuning methods with shared Adapter modules (Yang et al., 2024; Yao et al., 2025). With the advent of more powerful generative models, attention shifted toward synthesizing additional data to complement real few-shot samples, forming a joint data pool for model fine-tuning. CaFo (Zhang et al., 2023) enhances diversity of training dataset by combining prior knowledge of four large pretrained models.

The central challenge for this approach lies in guiding the synthesis process to generate data closely aligned with the real few-shot samples. IsSynth (He et al., 2023)

generates images by adding noise to real samples, while DISEF (da Costa et al., 2023) extends this approach by promoting diversity through image captioning and fine-tune CLIP models using LoRA (Hu et al., 2022) to reduce computational overhead. Both methods employ CLIP filtering to remove incorrectly classified samples. DataDream (Kim et al., 2024) advances this line of work by fine-tuning the generator with few-shot data, further aligning synthesized images with the real data distribution.

# 3. The Theoretical Benefits of Synthetic Data for Few-Shot Models

In this section, we theoretically analyze the role of synthetic data when being used to train a predictor. We derive a novel bound on the test error of a model, that explicitly reveals the role of a synthetic distribution, the discrepancy of the true and synthetic distribution, and the robustness of a predictor around the training (real and synthesized) samples. This bound provides novel insights and idea to train a predictor with synthetic data.

## 3.1. Preliminaries

*Notations:* A bold character (e.g., $z$) denotes a vector, while a bold capital (e.g., $S$) denotes a set. We denote by $\|\cdot\|$ the $\ell_2$-norm. $|S|$ denotes the size/cardinality of $S$, and $[K]$ denotes the set $\{1, 2, ..., K\}$ for $K \geq 1$. We will work with a model (or hypothesis) class $\mathcal{H}$, an instance set $\mathcal{Z}$, and a loss function $\ell : \mathcal{H} \times \mathcal{Z} \to \mathbb{R}$. Given a distribution $P$ defined on $\mathcal{Z}$, the quality of a model $h \in \mathcal{H}$ can be measured by its *expected loss* $F(P, h) = \mathbb{E}_{z \sim P}[\ell(h, z)]$. In practice, we typically collect a training set $S = \{z_1, ..., z_n\} \subseteq \mathcal{Z}$ and work with the *empirical loss* $F(S, h) = \frac{1}{|S|} \sum_{z \in S} \ell(h, z)$.

*Problem setting:* Let $S$ denotes a real dataset consisting of $n$ independent and identically distributed samples from the true data distribution $P_0$. Denote $\mathcal{G}$ as a generator that induces a synthetic data distribution $P_g$. We can use $\mathcal{G}$ to generate a synthetic dataset $G$ with $g$ samples. We are interested in training a classification model from the union $S \cup G$ of the real and synthetic datasets. This training problem is particularly important in many applications, especially for the cases that only few real samples can be collected.

To understand the role of synthetic data for training a predictor, we will use the following concepts:

**Definition 3.1** (Model-based discrepancy). Let $S$ and $G$ be two datasets and $h$ be a predictor. The $h$-based discrepancy between $S$ and $G$ is denoted as $\bar{d}_h(G, S) = \frac{1}{|G|.|S|} \sum_{u \in G, s \in S} \|h(s) - h(u)\|$.

This quantity can be seen as the model-dependent distance (through $h$) between two sets of real and synthetic samples $(G, S)$. One can easily extend this concept to measure the

discrepancy between a real distribution $P_0$ and a synthetic one $P_g$. Let $\bar{d}_h(P_g, P_0) = \mathbb{E}_{u \sim P_g, s \sim P_0} \|h(u) - h(s)\|$ be *the $h$-based discrepancy for the whole data space*, and $\bar{d}_h(P_g, P_0 | \mathcal{A}) = \mathbb{E}_{u \sim P_g, s \sim P_0}[\|h(u) - h(s)\| : u, s \in \mathcal{A}]$ be *the discrepancy for a local area $\mathcal{A}$*.

This concept is closely related to distribution matching, which has become a prominent technique for generating informative data for supervised learning. The idea of distribution matching has been applied extensively on dataset distillation (Zhao & Bilen, 2021; Zhao et al., 2023) where we generate a small dataset from a larger one, and has been used in large-scale image classification (Yuan et al., 2024).

Let $\Gamma(\mathcal{Z}) := \bigcup_{i=1}^{K} \mathcal{Z}_i$ be a partition of $\mathcal{Z}$ into $K$ disjoint nonempty subsets. Denote $T_S = \{i \in [K] : S \cap \mathcal{Z}_i \neq \emptyset\}$. In more details, $T_S$ is the collection of all valid areas (i.e., local areas which contain real data samples from $S$). Denote $S_i = S \cap \mathcal{Z}_i$, and $n_i = |S_i|$ as the number of samples falling into $\mathcal{Z}_i$, meaning that $n = \sum_{j=1}^{K} n_j$.

**Definition 3.2.** The *local robustness* of a predictor $h$ at a data instance $s \sim P$ in the area $\mathcal{A}$ is defined as $\mathcal{R}_h(s, \mathcal{A}|P) = \mathbb{E}_{z \sim P}[\|h(z) - h(s)\| : z \in \mathcal{A}]$.

By definition, $\mathcal{R}_h$ measures how robust a model is at a specific data point. A small local robustness suggests that the model should be robust in a small area around a point. Otherwise, the model may not be robust. Note that this definition uses the outputs from a model to define robustness.

We denote $\mathcal{R}_h(S, \mathcal{Z}_i|P_0) = \frac{1}{|S|} \sum_{s \in S} \mathcal{R}_h(s, \mathcal{Z}_i|P_0)$ and $\mathcal{R}_h(G, \mathcal{Z}_i|P_g) = \frac{1}{|G|} \sum_{g \in G} \mathcal{R}_h(g, \mathcal{Z}_i|P_g)$ as the local robustness of model $h$ on a real dataset $S$ and synthetic dataset $G$, respectively. Those quantities can be understood as a measurement of a model robustness with respect to a dataset.

## 3.2. Main Theorem

Given these two concepts, we now have the technical tools to provide a theoretical analysis for generalization ability of a model $h$ trained on both real and synthetic data ($S \cup G$). The following theorem presents an upper bound on the test error of a model, whose proof appears in Appendix A.1.

**Theorem 3.3.** *Consider a model $h$, a dataset $S$ containing $n$ i.i.d. samples from a real distribution $P_0$, and a synthetic dataset $G$ which contains i.i.d. samples from distribution $P_g$, so that $g_i = |G_i| > 0$ for each $i \in T_S$, where $G_i = G \cap \mathcal{Z}_i$. Let $C_h = \sup_{z \in \mathcal{Z}} \ell(h, z)$, $g = \sum_{i \in T_S} g_i$. Assume that the loss function $\ell(h, z)$ is $L_h$-Lipschitz continuous w.r.t $h$. For any $\delta > 0$, with probability at least $1 - \delta$, we have:*

$$F(P_0, h) \leq L_h \sum_{i \in T_S} \frac{g_i}{g} \left[ \bar{d}_h(G_i, S_i) + \mathcal{R}_h(G_i, \mathcal{Z}_i \mid P_g) \right] + A \tag{1}$$

*where* $A = F(P_g, h) + \sum_{i \in T_S} \left[ \frac{n_i}{n} - \frac{g_i}{g} \right] F(S_i, h) +$

$$L_h \sum_{i \in \boldsymbol{T}_S} \frac{n_i}{n} \mathcal{R}_h(\boldsymbol{S}_i, \mathcal{Z}_i | P_0) + C_h(\tfrac{1}{\sqrt{n}} + \tfrac{1}{\sqrt{g}}) \sqrt{2K \ln 2 + 2 \ln \tfrac{2}{\delta}}.$$

This theorem implies that we can bound the population loss $F(P_0, \boldsymbol{h})$ with synthetic data. This bound demonstrates that synthetic data can be leveraged to enhance the model's overall generalization performance. In particular, various implications can be inferred from (1):

1. The discrepancy term $\sum_{i \in \boldsymbol{T}_S} \frac{g_i}{g} \bar{d}_h(\boldsymbol{G}_i, \boldsymbol{S}_i)$ theoretically reveals that *the quality of the synthetic samples plays a crucial role for $\boldsymbol{h}$*. When those synthetic samples are close to the real ones, each $\bar{d}_h(\boldsymbol{G}_i, \boldsymbol{S}_i)$ would be small, suggesting that those synthetic samples can help improve generalization ability of $\boldsymbol{h}$. Otherwise, if there are some samples of $\boldsymbol{G}$ that are far (different) from the real samples, the bound (1) will be high, and not be optimal to train $\boldsymbol{h}$.

2. *The local robustness of $\boldsymbol{h}$ is also important to the generalization ability of $\boldsymbol{h}$*. Indeed, a decrease in robustness quantities of synthetic and real data ($\mathcal{R}_h(\boldsymbol{G}_i, \mathcal{Z}_i \mid P_g)$ and $\mathcal{R}_h(\boldsymbol{S}_i, \mathcal{Z}_i | P_0)$) can lead to a decrease in the population loss, leading to better generalization. Furthermore, they also reveal a connection between the local behavior and the generalization capability of the model.

3. The upper bound (1) *suggests an amenable way to generate synthetic samples*. Assume that we only have the real samples $\boldsymbol{S}$ which may be small. We can use a generator $\mathcal{G}$ to generate a synthetic set $\boldsymbol{G}$ that minimizes the upper bound (1). This amounts to optimize the synthetic samples to minimize the test error of predictor $\boldsymbol{h}$. It suggests that the *generated samples not only need to be close to the real ones, but also need to ensure better robustness* of the trained model at different local areas of the input space.

4. Theorem 3.3 also indicates that, for strong generalization on unseen data, the model $\boldsymbol{h}$ must not only accurately predict both real and synthetic samples, but also keep the discrepancy between real and synthetic predictions low and maintain local robustness for both data types. This reveals novel ways to train $\boldsymbol{h}$ from both real and synthetic data.

*Remark* 3.1 (**Tightness**). Though providing interesting insights, our bound (1) is not very tight for some aspects. For instance, it is $O(\sqrt{K})$ which can be not optimal when $K$ is large. Luckily, the few-shot setting does not allow us to choose a large $K$, since bound (1) mostly concerns on areas containing real samples. Furthermore, for the extreme cases with only one real sample, the "local" behavior of model $\boldsymbol{h}$ cannot be captured in our bound anymore, and hence our bound can be loose. On the other hand, the sample complexity of our bound seems to be optimal for both real and synthetic data. The reason is that the error of the best model is at least $O(g^{-0.5}) + const$ for a hard learning problem, according to Theorem 8 in (Than et al., 2025).

## 3.3. Asymptotic Case

Next we consider the asymptotic cases, which can help us to understand the roles of increasing the size of synthetic datasets and the quality of synthetic distribution. The following theorem reveals some new insights.

**Theorem 3.4.** *Using the same notations and assumptions of Theorem 3.3, and let $p_i^g = P_g(\mathcal{Z}_i)$ as the measure of area $\mathcal{Z}_i$ according to distribution $P_g$, for each $i \in \boldsymbol{T}_S$. For any $\delta > 0$, with probability at least $1 - \delta$, we have:*

$$F(P_0, \boldsymbol{h}) \leq L_h \sum_{i \in \boldsymbol{T}_S} \left[ p_i^g \mathcal{R}_h(\boldsymbol{S}_i, \mathcal{Z}_i | P_g) + \frac{n_i}{n} \mathcal{R}_h(\boldsymbol{S}_i, \mathcal{Z}_i | P_0) \right] + A_1$$

*where $A_1 = F(P_g, \boldsymbol{h}) + \sum_{i \in \boldsymbol{T}_S} \left[ \frac{n_i}{n} - p_i^g \right] F(\boldsymbol{S}_i, \boldsymbol{h}) + \frac{C_h}{\sqrt{n}} \sqrt{2K \ln 2 - 2 \ln \delta}$.*

This result suggests that by using a sufficiently large number of synthetic samples to train $\boldsymbol{h}$, we are making model $\boldsymbol{h}$ locally robust (w.r.t. both distributions) in every region containing the real samples, and hence improving generalization ability of the trained models. It also further explains the benefits of scaling the number of synthetic samples that have been observed empirically in prior studies.

Finally, we consider *how well the quality of the synthetic distribution $P_g$ can estimate the quality of model $\boldsymbol{h}$*. To this end, we assume that $n \rightarrow \infty$. In this case, it is easy to see that $\mathcal{R}_h(\boldsymbol{S}_i, \mathcal{Z}_i | P_g) \rightarrow \bar{d}_h(P_0, P_g | \mathcal{Z}_i)$ and $\frac{n_i}{n} \rightarrow p_i = P_0(\mathcal{Z}_i)$. This suggests $\left[ \frac{n_i}{n} - p_i^g \right] F(\boldsymbol{S}_i, \boldsymbol{h}) \rightarrow \left[ p_i - p_i^g \right] F_i(P_0, \boldsymbol{h})$, where $F_i(P_0, \boldsymbol{h}) = \mathbb{E}_{\boldsymbol{z} \sim P_0}[\ell(\boldsymbol{h}, \boldsymbol{z}) : \boldsymbol{z} \in \mathcal{Z}_i]$ is the expected loss of $\boldsymbol{h}$ in the area $\mathcal{Z}_i$. Furthermore, $\mathcal{R}_h(\boldsymbol{S}_i, \mathcal{Z}_i | P_0) \rightarrow \bar{d}_h(P_0, P_0 | \mathcal{Z}_i)$ as $n \rightarrow \infty$, and $\bar{d}_h(P_0, P_g) = \sum_i p_i^g \bar{d}_h(P_0, P_g | \mathcal{Z}_i)$. Therefore, the following result is a consequence of Theorem 3.4.

**Corollary 3.5.** *Given the notations and assumptions from Theorem 3.4, we have $\sum_{i=1}^K p_i^g F_i(P_0, \boldsymbol{h}) \leq F(P_g, \boldsymbol{h}) + L_h \sum_{i=1}^K \left[ p_i^g \bar{d}_h(P_0, P_g | \mathcal{Z}_i) + p_i \bar{d}_h(P_0, P_0 | \mathcal{Z}_i) \right]$. Moreover,*

$$\sum_{i=1}^K p_i^g F_i(P_0, \boldsymbol{h}) \leq F(P_g, \boldsymbol{h}) + L_h \left[ \bar{d}_h(P_0, P_g) + \bar{d}_h(P_0, P_0) \right]$$

This corollary provides a theoretical support for the intuition that a better synthetic distribution $P_g$ should be closer to the true one $P_0$. This can be derived from the discrepancy $\bar{d}_h(P_0, P_g)$ in our bound: The smaller this discrepancy is, the tighter bound on the test error of $\boldsymbol{h}$ we will get. Note that the left-hand side of the bound in Corollary 3.5 represents the macro-level average loss of $\boldsymbol{h}$.

It is worth highlighting an important implication of Corollary 3.5: **it suffices for the model $\boldsymbol{h}$ to perceive a small distance between $P_0$ and $P_g$.** This perspective differs fundamentally from traditional assumptions, which require the two distributions to be objectively close in some metric space. Our result suggests that even if $P_0$ and $P_g$ are far

apart in reality, strong generalization can still be achieved as long as the model $\boldsymbol{h}$ treats them as similar and incurs a low synthetic loss.

## 4. Algorithmic Design

This section presents how to use our theoretical bound on the generalization error to design an efficient algorithm to train a few-shot model, using synthetic data. We first discuss the main idea, and then the implementation details. While our discussion here focuses on classification problems, we believe that it is general enough to be applied to many other settings including regression.

### 4.1. Minimizing the Generalization Bound

To guarantee the performance of models trained with synthetic data on real data distribution to be small, we aim to minimize the R.H.S. of the bound (1). If we use a Lipschitz loss function for supervised learning problem such as absolute loss, the Lipschitz constant $L_h$ will not change. So we can ignore it from the bound above (e.g., by rescaling) from optimization perspective. For the sake of clarity, we rewrite the bound as the sum of the following terms:

$$A_1 = \sum_{i \in \boldsymbol{T}_S} \frac{g_i}{g} \bar{d}_h(\boldsymbol{G}_i, \boldsymbol{S}_i) \tag{2}$$

$$A_2 = \sum_{i \in \boldsymbol{T}_S} [\frac{g_i}{g} \mathcal{R}_h(\boldsymbol{G}_i, \mathcal{Z}_i | P_g) + \frac{n_i}{n} \mathcal{R}_h(\boldsymbol{S}_i, \mathcal{Z}_i | P_0)] \tag{3}$$

$$A_3 = F(\boldsymbol{S}, \boldsymbol{h}) - \sum_{i \in \boldsymbol{T}_S} \frac{g_i}{g} F(\boldsymbol{S}_i, \boldsymbol{h}) \tag{4}$$

$$A_4 = C_{\mathcal{H}}(n^{-0.5} + g^{-0.5})\sqrt{2K \ln 2 + 2 \ln(2/\delta)} \tag{5}$$

$$A_5 = F(P_g, \boldsymbol{h}) \tag{6}$$

Note that we use $\sum_{i \in \boldsymbol{T}_S} \frac{n_i}{n} F(\boldsymbol{S}_i, \boldsymbol{h}) = F(\boldsymbol{S}, \boldsymbol{h})$ here. Three components play a central role in our optimization problem: the partition $\Gamma$; the synthetic distribution $P_g$ and generated data $\boldsymbol{G}$; and the classifier $\boldsymbol{h}$. Because the generated data components will be partly addressed with a fine-tuning step for the generator (see Section 5.4 for empirical evidence), we focus on formalizing the other two.

#### 4.1.1. PARTITION OPTIMIZATION

In the previous bound, given a fixed real dataset $\boldsymbol{S}$ and a fixed generated dataset $\boldsymbol{G}$, we will minimize over the set of partitions $A_2$ (the quantity of distance between data point and its own regions) which depends on the partition the most in all of the above quantities. This optimization problem can be formalized as:

$$\min_{\Gamma(\mathcal{Z})} [\sum_{i \in \boldsymbol{T}_S} \frac{g_i}{g} \mathcal{R}_h(\boldsymbol{G}_i, \mathcal{Z}_i | P_g) + \sum_{i \in \boldsymbol{T}_S} \frac{n_i}{n} \mathcal{R}_h(\boldsymbol{S}_i, \mathcal{Z}_i | P_0)]$$

If we assume that in each region $i$, the amount of real and synthetic data ($n_i$ and $g_i$) are sufficiently large so that the

empirical distributions of the real and synthetic data can approximate well the true real and synthetic distributions $P_0$ and $P_g$ on that region, then the partition optimization objectives can be upper bounded by K-means clustering optimization objectives on prediction space. It serves as motivation for our choice of K-means clustering to solve the partitioning optimization. The proof appears in Appendix A.4.

#### 4.1.2. MODEL OPTIMIZATION

**Utilizing the few-shot learning setting**. The other component that heavily affects our generalization bound is the classifier $\boldsymbol{h}$ itself. Because the classifier affects all terms mentioned above except $A_4$, so we can formulate them as minimizing the sum of the remaining expression. Moreover, since we are focusing on few-shot real data, the intra-region distance term for real data ($\mathcal{R}_h(\boldsymbol{S}_i, \mathcal{Z}_i | P_0)$) could be ignored as it becomes negligibly small in this setting, and may not affect much to the model optimization.

By minimizing the classification losses on real and synthetic data ($A_3$ and $A_4$), we fine-tune the pretrained models on the target datasets. Optimizing the discrepancy term ($A_1$) then aligns the model's predictions between real and synthetic samples within each region, reducing any mismatch. Finally, minimizing the robustness term ($A_2$) ensures that the model maintains stable predictions on synthetic data, thereby enhancing overall predictive quality and improving generalization.

### 4.2. Algorithmic Details

We propose an algorithm (Algorithm 1) to address the two interrelated optimization problems discussed earlier through a two-phase optimization process. In the first phase, the algorithm optimizes data partitioning to address the initial optimization problem. In the second phase, it refines the classifiers to tackle the subsequent problem. This two-phase strategy is designed to minimize the loss function $\mathcal{L}$, which is formulated as a combination of distribution matching, robustness terms, and classification loss. Figure 1 visually demonstrates this general pipeline. The convergence of the loss function and the associated regularization terms is empirically demonstrated in Section 5.4.

**Loss function**:

$$\mathcal{L} = \lambda F(\boldsymbol{S}, \boldsymbol{h}) + F(\boldsymbol{G}, \boldsymbol{h})$$
$$+ \lambda_1 \sum_{i \in \boldsymbol{T}_S} \sum_{s \in \boldsymbol{S}_i, g \in \boldsymbol{G}_i} \frac{g_i}{g} \frac{1}{|\boldsymbol{G}_i||\boldsymbol{S}_i|} \|\boldsymbol{h}(s) - \boldsymbol{h}(g)\|$$
$$+ \lambda_2 \frac{1}{g} \sum_{i \in \boldsymbol{T}_S} \sum_{\boldsymbol{g}_1, \boldsymbol{g}_2 \in \boldsymbol{G}_i} \frac{1}{g_i} \|\boldsymbol{h}((\boldsymbol{g}_1) - \boldsymbol{h}(\boldsymbol{g}_2)\| \tag{7}$$

This loss function is directly inspired by the R.H.S. of Eq. 1

**Algorithm 1** Fine-tuning few-shot models with synthetic data

**Input**: Real dataset $S$, number $g$ of synthesis samples, (conditional) Pretrained generator models $\mathcal{G}$

 1: Initialize centroids $z$ for every local area
 2: Fine-tuning generator $\mathcal{G}$ by real dataset $S$ with LoRA
 3: Generate $g$ synthetic images from generator $\mathcal{G}$
 4: Use K-means clustering on both real and synthetic images to obtain partition $\Gamma(\mathcal{Z})$
 5:
 6: **for** each mini-batch $A$ **do**
 7:     Assign datapoints to their nearest clusters
 8:     Train the model $h$ using the loss function $\mathcal{L}$ on the combined dataset $S_A \cup G_A$ that includes both real data and synthetic data.     ▷ Refer to equation 7.
 9: **end for**

and model optimization problem. To minimize $A_3$ and $A_5$, we substitute them with the empirical loss on real and synthetic data. To address $A_1$ and $A_2$, we include them as regularization components in the loss function. Note that the expectation part in robustness term $A_2$ was calculated with its empirical version, utilizing other synthetic data in the same region to compute.

Our loss function is then defined as the sum of the empirical loss in real and synthetic data along with two regularization terms. We introduced hyper-parameters $\lambda$, $\lambda_1$, and $\lambda_2$ to control the influence of these terms in the optimization process. The empirical loss is modeled using the traditional cross-entropy loss, while the discrepancy and robustness terms are calculated using the $\ell_2$-norm to ensure smoother optimization.

We now describe the details of our algorithm:

**1. Initialization**: We begin with initializing the partitions and noise vectors essential for the iterations. Each iteration is one iteration of training the classifier. We fine-tune the generator with LoRA (Hu et al., 2022), with the same loss and procedure in DataDream (Kim et al., 2024). Then, we use this generator attached with LoRA module to generate synthetic data.

**2. Main Optimization process**:

- *Partition Optimization:* In part due to reasons mentioned in Section 4.1 and due to the simplicity of the K-means clustering algorithm, we decided to use it as a partition optimization algorithm. Furthermore, to save computation, we decided to perform clustering on data space avoid recomputing the clustering at each iteration.

- *Model Optimization:* With a stabilized partition, we optimize the classifier based on loss function $\mathcal{L}$. Forward real and synthetic data through models and compute necessary

components for loss computation (the terms $h(s)$ with real data $s \in S_i$, and $h(g)$ with synthetic data $g \in G_i$). These terms are computed as the softmax outputs of classification models.

- *Loss Computation and Gradient Descent:* We calculate the loss function $\mathcal{L}$ based on computed components. We update model $h$ via gradient descent.

Although this implementation achieves impressive results—thanks to the high-quality data produced by the fine-tuned generator—it also incurs computational costs, both for fine-tuning and for generating large volumes of synthetic data. As an alternative, we introduce a *lightweight* version that requires no generator fine-tuning phase and uses only about one-eighth as much synthetic data. The corresponding pseudocode and detailed description appear in Appendix C (Algorithm 2). As shown in the next section, even this streamlined approach provides competitive performance, compared to the state of the art.

## 5. Experiments

In this section, we validate the effectiveness of our proposed algorithm on few-shot image classification problems. First, we describe the experimental settings, including the baselines, datasets, and implementation details. Then, the main results of fine-tuning models are provided, followed by ablation studies and some additional analysis.

### 5.1. Experimental Settings

**Baselines.** We compared our solutions with other state-of-the-art methods in Few-shot Image classification: DataDream (Kim et al., 2024), DISEF (da Costa et al., 2023), and IsSynth (He et al., 2023). All of the results of the baseline methods were obtained from the DataDream paper, except for the DTD (Cimpoi et al., 2014) dataset, where we reproduced the results due to an erroneous implementation in their training/evaluation data split.

**Datasets.** Similar to baselines methods, we evaluate our method on 10 common datasets for few-shot image classification: FGVC Aircraft (Russakovsky et al., 2015) and Caltech101 (Li et al., 2022) for general object recognition, FGVC Aircraft (Maji et al., 2013) for fine-grained aircraft data, Food101 (Bossard et al., 2014) for common food objects, EuroSAT (Helber et al., 2019) for satellitle images, Oxford Pets (Parkhi et al., 2012) for discrimination of cat and dog types, DTD (Cimpoi et al., 2014) for texture images, SUN397 (Xiao et al., 2010) for scene understanding, Stanford Cars(Krause et al., 2013) for cars data, and Flowers102 (Nilsback & Zisserman, 2008) for flower classes.

**Experimental details.** We fine-tuned the CLIP ViT-B/16 image encoder with LoRA (Hu et al., 2022). To be consis-

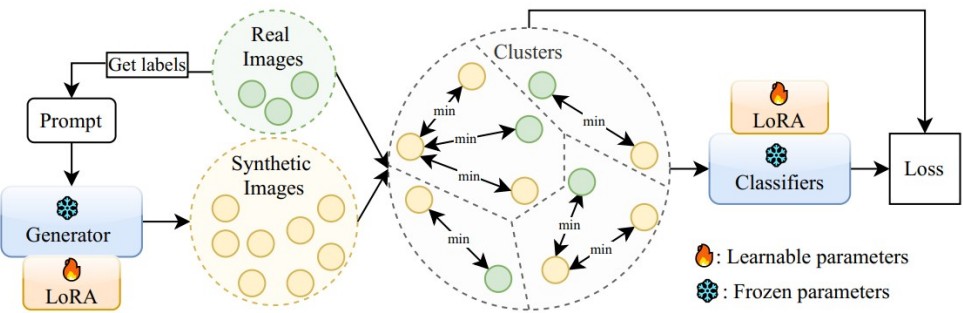

*Figure 1.* Illustration of the overall algorithm pipeline outlined in Algorithm 1. First, we generate synthetic images using the labels of real images. Subsequently, both real and synthetic images are clustered. Finally, the images are fed into the classifier. The final loss is calculated based on the model's predictions for the samples that belong to the same cluster, thereby reducing the prediction discrepancy between real and synthetic images and between synthetic images themselves within the same cluster.

*Table 1.* Few-shot image classification with CLIP ViT-B/16. All experiments are conducted with 16 real shots and 500 synthetic images per-class, except our lightweight version, where only 64 synthetic images per class were utilized. R/S columns denoted whether to use real or synthetic data, respectively. Each result of a method was averaged from 3 random seeds, except our full version, where we fixed the same seed 0 for all datasets.

| Method | R | S | IN | CAL | DTD | EuSAT | AirC | Pets | Cars | SUN | Food | FLO | Avg |
|---|---|---|---|---|---|---|---|---|---|---|---|---|---|
| CLIP (zero-shot) | | | 70.2 | 96.1 | 46.1 | 38.1 | 23.8 | 91.0 | 63.1 | 72.2 | 85.1 | 71.8 | 64.1 |
| Real-finetune | ✓ | | 73.4 | 96.8 | 73.9 | 93.5 | 59.3 | 94.0 | 87.5 | 77.1 | 87.6 | 98.7 | 84.2 |
| IsSynth | ✓ | ✓ | 73.9 | 97.4 | 75.1 | 93.9 | 64.8 | 92.1 | 88.5 | **77.7** | 86.0 | 99.0 | 84.8 |
| DISEF | ✓ | ✓ | 73.8 | 97.0 | 74.3 | 94.0 | 64.3 | 92.6 | 87.9 | 77.6 | 86.2 | 99.0 | 84.7 |
| DataDream$_{cls}$ | ✓ | ✓ | 73.8 | 97.6 | 73.1 | 93.8 | 68.3 | 94.5 | 91.2 | 77.5 | 87.5 | 99.4 | 85.7 |
| DataDream$_{dset}$ | ✓ | ✓ | **74.1** | 96.9 | 74.1 | 93.4 | 72.3 | **94.8** | 92.4 | 77.5 | 87.6 | **99.4** | 86.3 |
| Ours (lightweight) | ✓ | ✓ | 73.7 | **97.9** | **75.5** | 94.2 | 71.5 | 94.5 | 90.2 | 77.6 | 90.0 | 99.0 | 86.4 |
| Ours (full) | ✓ | ✓ | 73.8 | 97.3 | 74.5 | **94.7** | 74.3 | 94.6 | **93.1** | 77.7 | **90.4** | 99.3 | **87.0** |

tent with the baselines, the generator used is Stable Diffusion (SD) (Rombach et al., 2022) version 2.1. Similarly to the baselines, the guidance scale of SD is set to be 2.0 to enhance diversity. In the *lightweight* version, only 64 images per class were generated without the need to fine-tune the generator, and in the *full* version 500 images per class were synthesized from LoRA-attached fine-tuned Stable Diffusion. For the lightweight version, inspired by Real-Fake (Yuan et al., 2024), we improved the quality of the synthesized data using negative prompts `"distorted, unrealistic, blurry, out of frame"`.

The clustering phase was performed with the FAISS library (Douze et al., 2024). The hyperparameters to be tuned are: $\lambda_1, \lambda_2$ to control the discrepancy and robustness terms, number of clusters, learning rates, and weight decay. The values of $\lambda_1, \lambda_2$ vary between the data sets, but consistently maintain the ratio of 1/10, since we observe that this ratio brings the best balance between them and yields the best results. The hyperparameter $\lambda$ was chosen at 4 for all datasets except Stanford Cars, where we set it at 1. This choice resembles the choice in DataDream, where they select the weight for cross-entropy loss of real and synthetic data to be 0.8 and

0.2, respectively. For the number of clusters, we generally choose it twice as the number of classes of each dataset, except for ImageNet, where we set it to half of them. Analysis on the optimal choice of this hyperparameter can be found in the next section of Ablation Studies. More details of the hyperparameter settings can be found in Appendix B.

### 5.2. Main Results

Table 1 presents the main experimental results in ten datasets. Our method (in both lightweight and full variants) consistently outperforms existing state-of-the-art approaches, ranking first on 7 of the 10 datasets and second on 2 others. In the datasets where we do not achieve the top score, our results are within 0.1–0.3% of the best-performing method. On average, our lightweight variant performs on par with the strongest baseline (DataDream), while our full variant surpasses it by an additional 0.6%. The most notable gains are on datasets Food101 and challenging FGVC Aircraft, where our method improves performance by more than 2%.

*Table 2.* Ablation of the loss function components.

| Discre. | Rob. | EuroSAT | DTD | AirC | Cars |
|---------|------|---------|------|------|------|
|         |      | 93.5    | 74.1 | 72.5 | 92.6 |
|         | ✓    | 94.6    | 74.4 | 73.1 | **93.1** |
| ✓       |      | 94.3    | 74.3 | **74.8** | 93.0 |
| ✓       | ✓    | **94.7** | **74.5** | 74.3 | **93.1** |

*Table 3.* Methods performance on CLIP-Resnet50.

| Methods | AirC | Cars | Food | CAL |
|---------|------|------|------|-----|
| Real fine-tune | 61.57 | 78.86 | 63.52 | 93.29 |
| IsSynth | 70.94 | 90.82 | 68.77 | 94.54 |
| DISEF | 65.99 | 79.18 | 70.10 | 94.34 |
| DataDream$_{cls}$ | 79.21 | 92.99 | 66.70 | 94.37 |
| DataDream$_{dset}$ | 81.46 | 93.30 | 66.63 | **94.62** |
| Ours | **82.67** | **93.71** | **70.35** | 94.17 |

### 5.3. Ablation Studies

**Effectiveness of Regularization Terms**. We present results of ablation studies on the regularization terms in Table 2. There are four settings: no regularization, adding two terms of discrepancy and robustness subsequently, and adding both of them. As we can see from the results, adding the introduced regularization terms has positive effects on the results, with most of them have lead to increased performance.

**Analysis of partitioning**. We investigate how the number of clusters affects performance by varying it from a single cluster (where all data lie in one partition) to multiples of the total number of classes. We observe that the optimal choice is typically around twice the number of classes. We hypothesize that fewer clusters produce ambiguous decision boundaries, while too many clusters overly disperse the data, weakening regularization and degrading results. Based on these findings, we set the number of clusters to twice the number of classes in all main experiments, except for ImageNet, where we choose half of the classes as the number of clusters to reduce computational overhead. The detail of experiments can be found in Figure 3, Appendix D.

**Results on different architectures**. We provide additional results of our methods compared to other different methods on fine-tuning of the pre-trainedCLIP-Resnet50 in Table 3. We select 4 datasets of FGVC Aircraft, Stanford Cars, Food101, and Caltech101 as in additional experiments in (Kim et al., 2024). Our method outperforms others on 3 out of 4 datasets, while being competitive on the last one. This additional experiment shows the robustness of our method in different architectures, further demonstrating its superiority over existing current approaches.

### 5.4. Loss Convergence, Influence of Fine-Tuning, and Correlation with Discrepancy and Robustness

To further investigate the behavior of our algorithm and empirically validate the correctness of our generalization error bound, we compare four settings: using either our proposed loss function or the DataDream cross-entropy loss, each with or without a generator fine-tuning step. We focus on two challenging datasets: **DTD**, where fine-tuning the generator (as in DataDream and our full approach) unexpectedly degrades performance compared to methods without fine-tuning (IsSynth, DISEF, and our lightweight variant), and **FGVC Aircraft**, where Stable Diffusion is known to produce lower-quality images. We track the discrepancy and robustness terms (two regularization terms without multiplying by hyperparameters) throughout the training process of full version (measured twice per epoch for a total of 100 steps) and record the accuracy of each method. Our key observations from Figure 2 are:

- **Effective optimization and performance gains.** Our proposed algorithm successfully minimizes both the discrepancy and robustness terms, showing smoother and lower values compared to settings that do not incorporate these terms. Moreover, it achieves superior accuracy in both scenarios (with and without generator fine-tuning).

- **Impact of generator fine-tuning.** Fine-tuning the generator can substantially boost performance when the initial generative model is poorly suited to the domain (e.g., improving accuracy by up to 7% on FGVC Aircraft). Across most other datasets, methods that include this fine-tuning step (DataDream and our full version) also demonstrate improved results comparing to ones without them. Results validate our claim that fine-tuning can partially reduce the discrepancy and robustness terms in Section 4.1.

- **Empirical support for our theoretical framework.** Models that achieve smaller discrepancy and robustness values generally exhibit better accuracy, indicating that these metrics serve as reliable indicators of performance and generalization ability. Our ablations suggest that the robustness term, which has been overlooked in prior studies, seem to be really important to ensure high generalization of the trained model.

## 6. Conclusion

In this paper, we introduce a *theoretically guided* method for training few-shot models with synthetic data. We begin by deriving a generalization bound that reveals how the misalignment between real and synthetic data, as well as model robustness, affects performance and generalization. Building directly on these theoretical insights, we propose the first algorithm with theoretical guarantees to minimize

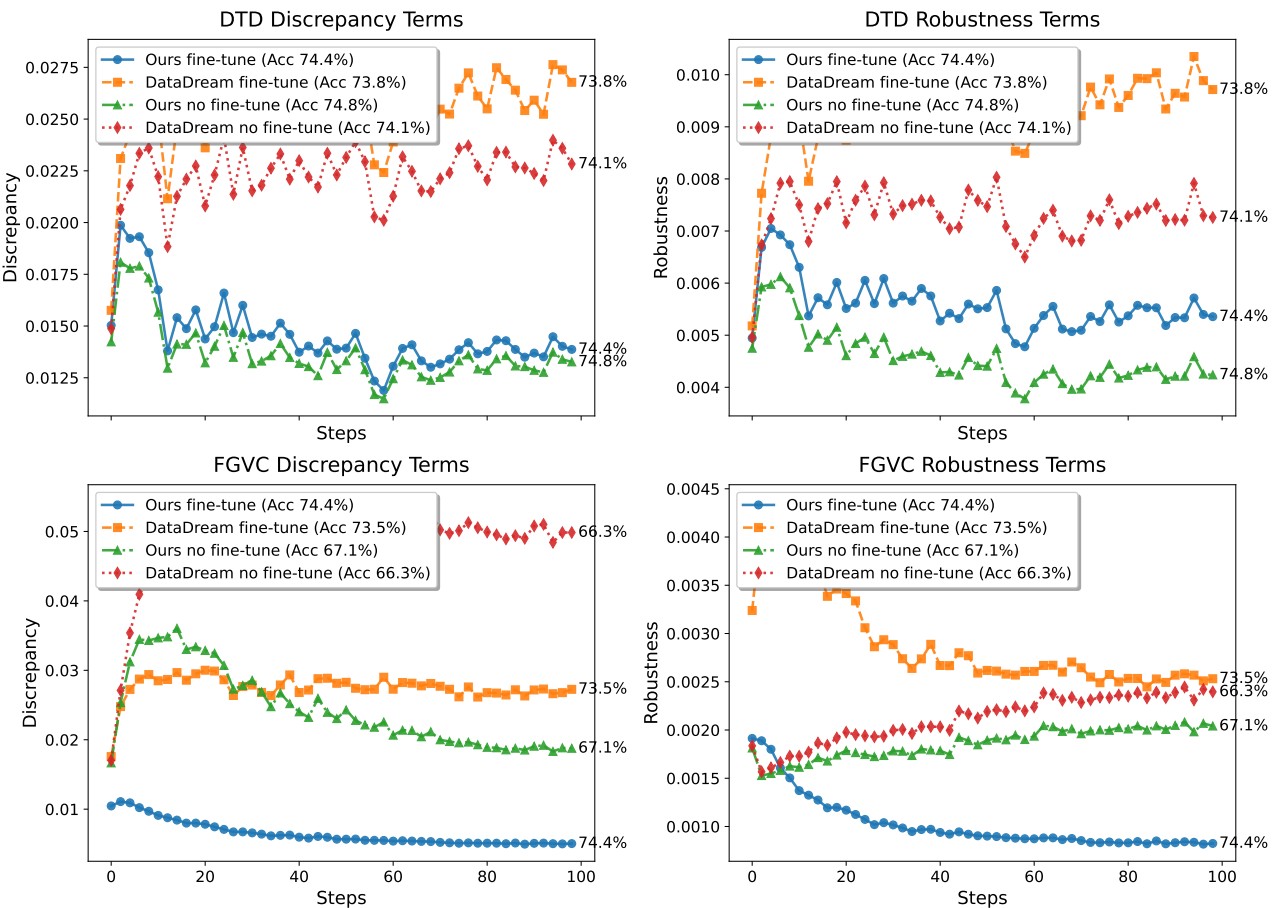

*Figure 2.* Visualization of Discrepancy and Robustness Terms Across Settings for DTD and FGVC Aircraft Datasets. The top row shows results for the DTD dataset, with discrepancy (left) and robustness (right) terms, while the bottom row shows results for the FGVC Aircraft dataset. Accuracy values are annotated at the end of each corresponding line.

this bound, thereby maximizing the performance of few-shot models. Extensive experiments on ten benchmark datasets demonstrate that our method consistently outperforms the state-of-the-art.

Future works can come from additional analysis to fully solve the partition and model selection problems, and find an efficient way to directly use this theoretical framework for optimizing synthetic data through fine-tuning generator or filtering by using our loss function as a criterion. Moreover, thanks to the versatility and simplicity of this generalization bound, one can use them for enhancing performance of related domains such as adversarial training or domain adaptation with or without additional synthetic data.

## Impact Statement

This work aims to leverage synthetic data for improving few-shot classification models, which holds promise for expanding machine learning applications in real-world scenarios with limited labeled data. In most cases, by reducing reliance on large-scale annotated datasets, our approach can potentially democratize access to high-performance models. However, when dealing with synthetic data, the method, if being misused, may facilitate malicious activities such as model poisoning or creation of deceptive content. Furthermore, part of the ideas can be extended to related topics in Machine Learning such as adversarial training and domain adaptation, thus sharing the same societal impact on those fields.

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

## A. Proofs

### A.1. Proof of Theorem 3.3

*Proof.* In the following analysis, we will denote $F(\emptyset, \boldsymbol{h}) = 0$. We first observe that:

$$
\begin{aligned}
F(P_0, \boldsymbol{h}) \quad = \quad & F(P_g, \boldsymbol{h}) + F(P_0, \boldsymbol{h}) - F(\boldsymbol{S}, \boldsymbol{h}) + F(\boldsymbol{S}, \boldsymbol{h}) - \sum_{i \in \boldsymbol{T}_S} \frac{g_i}{g} F(\boldsymbol{S}_i, \boldsymbol{h}) \\
& + \sum_{i \in \boldsymbol{T}_S} \frac{g_i}{g} F(\boldsymbol{S}_i, \boldsymbol{h}) - F(\boldsymbol{G}, \boldsymbol{h}) + F(\boldsymbol{G}, \boldsymbol{h}) - F(P_g, \boldsymbol{h})
\end{aligned} \tag{8}
$$

Next, we consider

$$
\begin{aligned}
\sum_{i \in \boldsymbol{T}_S} \frac{g_i}{g} F(\boldsymbol{S}_i, \boldsymbol{h}) - F(\boldsymbol{G}, \boldsymbol{h}) \quad &= \quad \sum_{i \in \boldsymbol{T}_S} \frac{g_i}{g} F(\boldsymbol{S}_i, \boldsymbol{h}) - \frac{1}{g} \sum_{i \in \boldsymbol{T}_S} \sum_{\boldsymbol{u} \in \boldsymbol{G}_i} \ell(\boldsymbol{h}, \boldsymbol{u}) \tag{9} \\
&= \quad \frac{1}{g} \sum_{i \in \boldsymbol{T}_S} \sum_{\boldsymbol{u} \in \boldsymbol{G}_i} [F(\boldsymbol{S}_i, \boldsymbol{h}) - \ell(\boldsymbol{h}, \boldsymbol{u})] \tag{10} \\
&= \quad \frac{1}{g} \sum_{i \in \boldsymbol{T}_S} \sum_{\boldsymbol{u} \in \boldsymbol{G}_i} \frac{1}{n_i} \sum_{\boldsymbol{s} \in \boldsymbol{S}_i} [\ell(\boldsymbol{h}, \boldsymbol{s}) - \ell(\boldsymbol{h}, \boldsymbol{u})] \tag{11} \\
&\leq \quad \frac{1}{g} \sum_{i \in \boldsymbol{T}_S} \sum_{\boldsymbol{u} \in \boldsymbol{G}_i} \frac{1}{n_i} \sum_{\boldsymbol{s} \in \boldsymbol{S}_i} L_h \|\boldsymbol{h}(\boldsymbol{s}) - \boldsymbol{h}(\boldsymbol{u})\| \tag{12} \\
&= \quad \frac{L_h}{g} \sum_{i \in \boldsymbol{T}_S} \sum_{\boldsymbol{u} \in \boldsymbol{G}_i} \frac{1}{n_i} \sum_{\boldsymbol{s} \in \boldsymbol{S}_i} \|\boldsymbol{h}(\boldsymbol{s}) - \boldsymbol{h}(\boldsymbol{u})\| \tag{13} \\
&= \quad \frac{L_h}{g} \sum_{i \in \boldsymbol{T}_S} \sum_{\boldsymbol{u} \in \boldsymbol{G}_i} \bar{d}_h(\boldsymbol{u}, \boldsymbol{S}_i) \tag{14} \\
&= \quad L_h \sum_{i \in \boldsymbol{T}_S} \frac{g_i}{g} \bar{d}_h(\boldsymbol{G}_i, \boldsymbol{S}_i) \tag{15}
\end{aligned}
$$

where we have used the fact that $\ell(\boldsymbol{h}, \boldsymbol{s}) - \ell(\boldsymbol{h}, \boldsymbol{u}) \leq L_h \|\boldsymbol{h}(\boldsymbol{s}) - \boldsymbol{h}(\boldsymbol{u})\|$, due to the Lipschitz continuity of $\ell$.

By Theorem A.1, for any $\delta > 0$, we have each of the followings with probability at least $1 - \delta/2$:

$$
F(\boldsymbol{G}, \boldsymbol{h}) - F(P_g, \boldsymbol{h}) \quad \leq \quad L_h \sum_{i \in \boldsymbol{T}_S} \frac{g_i}{g} \mathcal{R}_h(\boldsymbol{G}_i, \mathcal{Z}_i | P_g) + C \sqrt{\frac{2K \ln 2 + 2 \ln(2/\delta)}{g}} \tag{16}
$$

$$
F(P_0, \boldsymbol{h}) - F(\boldsymbol{S}, \boldsymbol{h}) \quad \leq \quad L_h \sum_{i \in \boldsymbol{T}_S} \frac{n_i}{n} \mathcal{R}_h(\boldsymbol{S}_i, \mathcal{Z}_i | P_0) + C \sqrt{\frac{2K \ln 2 + 2 \ln(2/\delta)}{n}} \tag{17}
$$

Combining (8) with (15,16,17) will complete the proof. $\qquad\square$

### A.2. Proof of Theorem 3.4

*Proof.* Using the same arguments as before, we first observe that:

$$
F(P_0, \boldsymbol{h}) \quad = \quad F(P_0, \boldsymbol{h}) - F(\boldsymbol{S}, \boldsymbol{h}) + F(\boldsymbol{S}, \boldsymbol{h}) - \sum_{i \in \boldsymbol{T}_S} \frac{g_i}{g} F(\boldsymbol{S}_i, \boldsymbol{h}) + \sum_{i \in \boldsymbol{T}_S} \frac{g_i}{g} F(\boldsymbol{S}_i, \boldsymbol{h}) - F(\boldsymbol{G}, \boldsymbol{h}) + F(\boldsymbol{G}, \boldsymbol{h}) \tag{18}
$$

Next, we consider

$$\sum_{i \in \boldsymbol{T}_S} \frac{g_i}{g} F(\boldsymbol{S}_i, \boldsymbol{h}) - F(\boldsymbol{G}, \boldsymbol{h}) \quad = \quad \sum_{i \in \boldsymbol{T}_S} \frac{g_i}{g} F(\boldsymbol{S}_i, \boldsymbol{h}) - \frac{1}{g} \sum_{i \in \boldsymbol{T}_S} \sum_{\boldsymbol{u} \in \boldsymbol{G}_i} \ell(\boldsymbol{h}, \boldsymbol{u}) \tag{19}$$

$$= \quad \frac{1}{g} \sum_{i \in \boldsymbol{T}_S} \sum_{\boldsymbol{u} \in \boldsymbol{G}_i} [F(\boldsymbol{S}_i, \boldsymbol{h}) - \ell(\boldsymbol{h}, \boldsymbol{u})] \tag{20}$$

$$= \quad \frac{1}{g} \sum_{i \in \boldsymbol{T}_S} \sum_{\boldsymbol{u} \in \boldsymbol{G}_i} \frac{1}{n_i} \sum_{\boldsymbol{s} \in \boldsymbol{S}_i} [\ell(\boldsymbol{h}, \boldsymbol{s}) - \ell(\boldsymbol{h}, \boldsymbol{u})] \tag{21}$$

$$\leq \quad \frac{1}{g} \sum_{i \in \boldsymbol{T}_S} \sum_{\boldsymbol{u} \in \boldsymbol{G}_i} \frac{1}{n_i} \sum_{\boldsymbol{s} \in \boldsymbol{S}_i} L_h \|\boldsymbol{h}(\boldsymbol{s}) - \boldsymbol{h}(\boldsymbol{u})\| \tag{22}$$

$$= \quad \frac{L_h}{g} \sum_{i \in \boldsymbol{T}_S} \sum_{\boldsymbol{u} \in \boldsymbol{G}_i} \frac{1}{n_i} \sum_{\boldsymbol{s} \in \boldsymbol{S}_i} \|\boldsymbol{h}(\boldsymbol{s}) - \boldsymbol{h}(\boldsymbol{u})\| \tag{23}$$

$$= \quad \frac{L_h}{g} \sum_{i \in \boldsymbol{T}_S} \sum_{\boldsymbol{u} \in \boldsymbol{G}_i} \bar{d}_h(\boldsymbol{u}, \boldsymbol{S}_i) \tag{24}$$

$$= \quad L_h \sum_{i \in \boldsymbol{T}_S} \frac{g_i}{g} \bar{d}_h(\boldsymbol{G}_i, \boldsymbol{S}_i) \tag{25}$$

where we have used the fact that $\ell(\boldsymbol{h}, \boldsymbol{s}) - \ell(\boldsymbol{h}, \boldsymbol{u}) \leq L_h \|\boldsymbol{h}(\boldsymbol{s}) - \boldsymbol{h}(\boldsymbol{u})\|$, due to the Lipschitz continuity of $\ell$.

By Theorem A.1, for any $\delta > 0$, we have the following with probability at least $1 - \delta$:

$$F(P_0, \boldsymbol{h}) - F(\boldsymbol{S}, \boldsymbol{h}) \quad \leq \quad L_h \sum_{i \in \boldsymbol{T}_S} \frac{n_i}{n} \mathcal{R}_h(\boldsymbol{S}_i, \mathcal{Z}_i | P_0) + C \sqrt{\frac{2K \ln 2 + 2 \ln(1/\delta)}{n}} \tag{26}$$

Combining (18) with (25,26), we have the following with probability at least $1 - \delta$:

$$F(P_0, \boldsymbol{h}) \quad \leq \quad L_h \sum_{i \in \boldsymbol{T}_S} \frac{n_i}{n} \mathcal{R}_h(\boldsymbol{S}_i, \mathcal{Z}_i | P_0) + C \sqrt{\frac{2K \ln 2 + 2 \ln(1/\delta)}{n}} + F(\boldsymbol{S}, \boldsymbol{h}) - \sum_{i \in \boldsymbol{T}_S} \frac{g_i}{g} F(\boldsymbol{S}_i, \boldsymbol{h})$$

$$+ L_h \sum_{i \in \boldsymbol{T}_S} \frac{g_i}{g} \bar{d}_h(\boldsymbol{G}_i, \boldsymbol{S}_i) + F(\boldsymbol{G}, \boldsymbol{h}) \tag{27}$$

As $g \to \infty$, observe that $\bar{d}_h(\boldsymbol{G}_i, \boldsymbol{S}_i) \to \mathcal{R}_h(\boldsymbol{S}_i, \mathcal{Z}_i | P_g)$ and $F(\boldsymbol{G}, \boldsymbol{h}) \to F(P_g, \boldsymbol{h})$ and $\frac{g_i}{g} \to p_i^g$. Combining those facts with (27) completes the proof. $\square$

## A.3. Some necessary bounds

**Theorem A.1.** *Consider a model $\boldsymbol{h}$ learned from a dataset $\boldsymbol{S}$ with $n$ i.i.d. samples from distribution $P$. Let $C = \sup_{\boldsymbol{z} \in \mathcal{Z}} \ell(\boldsymbol{h}, \boldsymbol{z})$. Assume that the loss function $\ell(\boldsymbol{h}, \boldsymbol{z})$ is $L_h$-Lipschitz continuous w.r.t $\boldsymbol{h}$. For any $\delta > 0$, the following holds with probability at least $1 - \delta$:*

$$F(P, \boldsymbol{h}) \quad \leq \quad F(\boldsymbol{S}, \boldsymbol{h}) + L_h \sum_{i \in \boldsymbol{T}} \frac{n_i}{n} \mathcal{R}_h(\boldsymbol{S}_i, \mathcal{Z}_i | P) + C \sqrt{\frac{2K \ln 2 - 2 \ln \delta}{n}} \tag{28}$$

$$F(\boldsymbol{S}, \boldsymbol{h}) \quad \leq \quad F(P, \boldsymbol{h}) + L_h \sum_{i \in \boldsymbol{T}} \frac{n_i}{n} \mathcal{R}_h(\boldsymbol{S}_i, \mathcal{Z}_i | P) + C \sqrt{\frac{2K \ln 2 - 2 \ln \delta}{n}} \tag{29}$$

*Proof.* Firstly, we make the following decomposition:

$$F(P, \boldsymbol{h}) = F(P, \boldsymbol{h}) - \sum_{i=1}^{K} \frac{n_i}{n} \mathbb{E}_{\boldsymbol{z} \sim P}[\ell(\boldsymbol{h}, \boldsymbol{z}) | \boldsymbol{z} \in \mathcal{Z}_i] \tag{30}$$

$$+ \sum_{i=1}^{K} \frac{n_i}{n} \mathbb{E}_{\boldsymbol{z} \sim P}[\ell(\boldsymbol{h}, \boldsymbol{z}) | \boldsymbol{z} \in \mathcal{Z}_i] - F(\boldsymbol{S}, \boldsymbol{h}) + F(\boldsymbol{S}, \boldsymbol{h}) \tag{31}$$

Secondly, observe that

$$
F(P, \boldsymbol{h}) - \sum_{i=1}^{K} \frac{n_i}{n} \mathbb{E}_{\boldsymbol{z} \sim P}[\ell(\boldsymbol{h}, \boldsymbol{z}) | \boldsymbol{z} \in \mathcal{Z}_i] = \sum_{i=1}^{K} P(\mathcal{Z}_i) \mathbb{E}_{\boldsymbol{z} \sim P}[\ell(\boldsymbol{h}, \boldsymbol{z}) | \boldsymbol{z} \in \mathcal{Z}_i]
$$

$$
- \sum_{i=1}^{K} \frac{n_i}{n} \mathbb{E}_{\boldsymbol{z} \sim P}[\ell(\boldsymbol{h}, \boldsymbol{z}) | \boldsymbol{z} \in \mathcal{Z}_i] \tag{32}
$$

$$
= \sum_{i=1}^{K} \mathbb{E}_{\boldsymbol{z} \sim P}[\ell(\boldsymbol{h}, \boldsymbol{z}) | \boldsymbol{z} \in \mathcal{Z}_i] \left[ P(\mathcal{Z}_i) - \frac{n_i}{n} \right] \tag{33}
$$

$$
\leq C \sum_{i=1}^{K} \left| P(\mathcal{Z}_i) - \frac{n_i}{n} \right| \tag{34}
$$

Note that $(n_1, ..., n_K)$ is an i.i.d multinomial random variable with parameters $n$ and $(P(\mathcal{Z}_1), ..., P(\mathcal{Z}_K))$. Bretagnolle–Huber–Carol inequality, shows the following for any $\epsilon > 0$:

$$
\Pr \left( \sum_{i=1}^{K} \left| P(\mathcal{Z}_i) - \frac{n_i}{n} \right| \geq 2\epsilon \right) \leq 2^n \exp(-2n\epsilon^2)
$$

In other words, for any $\delta > 0$, the following holds true with probability at least $1 - \delta$:

$$
\begin{aligned}
F(P, \boldsymbol{h}) - \sum_{i=1}^{K} \frac{n_i}{n} \mathbb{E}_{\boldsymbol{z} \sim P}[\ell(\boldsymbol{h}, \boldsymbol{z}) | \boldsymbol{z} \in \mathcal{Z}_i] & \leq C \sum_{i=1}^{K} \left| P(\mathcal{Z}_i) - \frac{n_i}{n} \right| \\
& \leq C \sqrt{\frac{2K \ln 2 - 2 \ln \delta}{n}}
\end{aligned} \tag{35}
$$

Furthermore,

$$
\begin{aligned}
\sum_{i=1}^{K} \frac{n_i}{n} \mathbb{E}_{\boldsymbol{z} \sim P}[\ell(\boldsymbol{h}, \boldsymbol{z}) | \boldsymbol{z} \in \mathcal{Z}_i] - F(\boldsymbol{S}, \boldsymbol{h}) & = \sum_{i=1}^{K} \frac{n_i}{n} \mathbb{E}_{\boldsymbol{z} \sim P}[\ell(\boldsymbol{h}, \boldsymbol{z}) | \boldsymbol{z} \in \mathcal{Z}_i] - \frac{1}{n} \sum_{\boldsymbol{s} \in \boldsymbol{S}} \ell(\boldsymbol{h}, \boldsymbol{s}) \tag{36} \\
& = \sum_{i \in \boldsymbol{T}_S} \frac{n_i}{n} \mathbb{E}_{\boldsymbol{z} \sim P}[\ell(\boldsymbol{h}, \boldsymbol{z}) | \boldsymbol{z} \in \mathcal{Z}_i] - \frac{1}{n} \sum_{i \in \boldsymbol{T}_S} \sum_{\boldsymbol{s} \in \boldsymbol{S}_i} \ell(\boldsymbol{h}, \boldsymbol{s}) \tag{37} \\
& = \frac{1}{n} \sum_{i \in \boldsymbol{T}_S} \left[ n_i \mathbb{E}_{\boldsymbol{z} \sim P}[\ell(\boldsymbol{h}, \boldsymbol{z}) | \boldsymbol{z} \in \mathcal{Z}_i] - \sum_{\boldsymbol{s} \in \boldsymbol{S}_i} \ell(\boldsymbol{h}, \boldsymbol{s}) \right] \tag{38} \\
& = \frac{1}{n} \sum_{i \in \boldsymbol{T}_S} \sum_{\boldsymbol{s} \in \boldsymbol{S}_i} \mathbb{E}_{\boldsymbol{z} \sim P}[\ell(\boldsymbol{h}, \boldsymbol{z}) - \ell(\boldsymbol{h}, \boldsymbol{s}) : \boldsymbol{z} \in \mathcal{Z}_i] \tag{39} \\
& \leq \frac{1}{n} \sum_{i \in \boldsymbol{T}_S} \sum_{\boldsymbol{s} \in \boldsymbol{S}_i} \mathbb{E}_{\boldsymbol{z} \sim P}[L_h \| \boldsymbol{h}(\boldsymbol{z}) - \boldsymbol{h}(\boldsymbol{s}) \| : \boldsymbol{z} \in \mathcal{Z}_i] \tag{40} \\
& = \frac{L_h}{n} \sum_{i \in \boldsymbol{T}_S} \sum_{\boldsymbol{s} \in \boldsymbol{S}_i} \mathbb{E}_{\boldsymbol{z} \sim P}[\| \boldsymbol{h}(\boldsymbol{z}) - \boldsymbol{h}(\boldsymbol{s}) \| : \boldsymbol{z} \in \mathcal{Z}_i] \tag{41} \\
& = L_h \sum_{i \in \boldsymbol{T}} \frac{n_i}{n} \mathcal{R}_h(\boldsymbol{S}_i, \mathcal{Z}_i | P) \tag{42}
\end{aligned}
$$

Where (40) comes from (39) since $\ell$ is $L_h$-Lipschitz continuous w.r.t $\boldsymbol{h}$. Combining the decomposition (30) with (35) and (42) will arrive at (28). One can use the same arguments as above with reverse order of empirical and population loss to show (29), completing the proof. $\square$

## A.4. Proof of property in partition optimization 4.1.1

**Lemma A.2.** *If $\boldsymbol{x_i}$ and $\boldsymbol{x_j}$ are vectors from the same clusters, $\bar{x}$ is the mean of that cluster, and $n$ is the number of data points in that cluster we have the following expression*

$$\sum_{i,j} \|\boldsymbol{x_i} - \boldsymbol{x_j}\|^2 = \sum_{i \neq j} \|(\boldsymbol{x_i} - \bar{\boldsymbol{x}}) - (\boldsymbol{x_j} - \bar{\boldsymbol{x}})\|^2 = 2n \sum_{i=1}^{n} \|\boldsymbol{x_i} - \bar{\boldsymbol{x}}\|^2$$

**Proof:** If we extract the inside of the 2nd expression above:

$$\|(\boldsymbol{x_i} - \bar{\boldsymbol{x}}) - (\boldsymbol{x_j} - \bar{\boldsymbol{x}})\|^2 = \|\boldsymbol{x_i} - \bar{\boldsymbol{x}}\|^2 + \|\boldsymbol{x_j} - \bar{\boldsymbol{x}}\|^2 - 2(\boldsymbol{x_i} - \bar{\boldsymbol{x}})^T (\boldsymbol{x_j} - \bar{\boldsymbol{x}})$$

Due to symmetry, we can evaluate the first two expressions easily:

$$\sum_{i \neq j} \|\boldsymbol{x_i} - \bar{\boldsymbol{x}}\|^2 = \sum_{i \neq j} \|\boldsymbol{x_j} - \bar{\boldsymbol{x}}\|^2 = (n-1) \sum_{i=1}^{n} \|\boldsymbol{x_i} - \bar{\boldsymbol{x}}\|^2$$

The last expression is:

$$\left(\text{Note that } \sum_{j \neq i}(\boldsymbol{x_j} - \bar{\boldsymbol{x}}) = \sum_{j=1}^{n}(\boldsymbol{x_j} - \bar{\boldsymbol{x}}) - (\boldsymbol{x_i} - \bar{\boldsymbol{x}}) = n\bar{\boldsymbol{x}} - n\bar{\boldsymbol{x}} - (\boldsymbol{x_i} - \bar{\boldsymbol{x}}) = -(\boldsymbol{x_i} - \bar{\boldsymbol{x}})\right)$$

$$\sum_{i \neq j} -2(\boldsymbol{x_i} - \bar{\boldsymbol{x}})^T (\boldsymbol{x_j} - \bar{\boldsymbol{x}}) = -2 \sum_{i=1}^{n}(\boldsymbol{x_i} - \bar{\boldsymbol{x}})^T \sum_{j \neq i}(\boldsymbol{x_j} - \bar{\boldsymbol{x}}) = 2 \sum_{i=1}^{n}(\boldsymbol{x_i} - \bar{\boldsymbol{x}})^T(-(\boldsymbol{x_i} - \bar{\boldsymbol{x}}))$$

$$= 2 \sum_{i=1}^{n} \|\boldsymbol{x_i} - \bar{\boldsymbol{x}}\|^2$$

Finally, combining all, we have $(2(n-1) + 2) \sum_{i=1}^{n} \|\boldsymbol{x_i} - \bar{\boldsymbol{x}}\|^2 = 2n \sum_{i=1}^{n} \|\boldsymbol{x_i} - \bar{\boldsymbol{x}}\|^2$

**Main Proof** Now, coming back to the main results, we rewrite here the partition optimization problem:

$$\min_{\Gamma(\mathcal{Z})} \left[ \sum_{i \in \boldsymbol{T_S}} \frac{g_i}{g} \mathcal{R}_h(\boldsymbol{G_i}, \mathcal{Z}_i | P_g) + \sum_{i \in \boldsymbol{T_S}} \frac{n_i}{n} \mathcal{R}_h(\boldsymbol{S_i}, \mathcal{Z}_i | P_0) \right] \tag{43}$$

$$= \min_{\Gamma(\mathcal{Z})} \left[ \sum_{i \in \boldsymbol{T_S}} \frac{g_i}{g} \frac{1}{g_i} \sum_{\boldsymbol{g} \in \boldsymbol{G_i}} \mathcal{R}_h(\boldsymbol{g}, \mathcal{Z}_i | P_g) + \sum_{i \in \boldsymbol{T_S}} \frac{n_i}{n} \frac{1}{n_i} \sum_{\boldsymbol{s} \in \boldsymbol{S_i}} \mathcal{R}_h(\boldsymbol{s}, \mathcal{Z}_i | P_0) \right] \tag{44}$$

$$= \min_{\Gamma(\mathcal{Z})} \left[ \frac{1}{g} \sum_{i \in \boldsymbol{T_S}} \sum_{\boldsymbol{g} \in \boldsymbol{G_i}} \mathbb{E}_{\boldsymbol{z} \sim P_g}[\|\boldsymbol{h}(\boldsymbol{z}) - \boldsymbol{h}(\boldsymbol{g})\| : \boldsymbol{z} \in \mathcal{Z}_i] + \frac{1}{n} \sum_{i \in \boldsymbol{T_S}} \sum_{\boldsymbol{s} \in \boldsymbol{S_i}} \mathbb{E}_{\boldsymbol{z} \sim P_0}[\|\boldsymbol{h}(\boldsymbol{z}) - \boldsymbol{h}(\boldsymbol{s})\| : \boldsymbol{z} \in \mathcal{Z}_i] \right] \tag{45}$$

Next we deal with each term in above expression separately. We approximate the expectation term by its empirical version as follows:

$$\approx \min_{\Gamma(\mathcal{Z})} \frac{1}{g} \sum_{i \in \boldsymbol{T_S}} \sum_{\boldsymbol{g} \in \boldsymbol{G_i}} \sum_{\boldsymbol{z} \in \boldsymbol{G_i}} \frac{1}{g_i} \|\boldsymbol{h}(\boldsymbol{z}) - \boldsymbol{h}(\boldsymbol{g})\| \tag{46}$$

For easier derivation, we will minimize the sum of squares of distance. Note that the square of the sum of positive number is always smaller than or equal to sum of their respective square multiply with a constant. So basically we are minimizing an upper bound of the problem. Denote $\mu_i$ as the average of all the output of generated samples in the region $i$ (average of all $\boldsymbol{h}(z)$ with $z \in \boldsymbol{G_i}$), and our optimization problem becomes:

$$\min_{\Gamma(\mathcal{Z})} \frac{1}{g} \sum_{i \in \boldsymbol{T_S}} \sum_{\boldsymbol{g} \in \boldsymbol{G_i}} \sum_{\boldsymbol{z} \in \boldsymbol{G_i}} \frac{1}{g_i} \|\boldsymbol{h}(\boldsymbol{z}) - \boldsymbol{h}(\boldsymbol{g})\|^2 \tag{47}$$

$$= \min_{\Gamma(\mathcal{Z})} \frac{2}{g} \sum_{i \in \boldsymbol{T_S}} \sum_{\boldsymbol{z} \in \boldsymbol{G_i}} \|\boldsymbol{h}(\boldsymbol{z}) - \mu_i\|^2 \tag{48}$$

Note that the quantity inside the sum of (47) is called the within-cluster variation of the K-means problem and is equivalent to the traditional variation in (48). The proof is provided in Lemma A.2. Our bound now becomes the lower bound of the K-means optimization problem, and if we ignore the term of summing only over the region containing real samples, it becomes a K-means problem and can be solved easily. The same argument can be applied for the second term.

## B. Hyperparameters settings

We adopt the same data preprocessing pipeline as in the baseline DataDream (Kim et al., 2024), applying standard augmentations including random horizontal flipping, random resized cropping, random color jitter, random grayscale, Gaussian blur, and solarization. Our main difference is to use CutMix (Yun et al., 2019) and Mixup (Zhang et al., 2018) by default on all datasets to reduce the amount of hyperparameter tuning. We train our models using AdamW (Loshchilov & Hutter, 2019), searching the learning rate in $\{2e{-}4, 1e{-}4, 1e{-}5, 1e{-}6\}$ and the weight decay in $\{1e{-}3, 5e{-}4, 1e{-}4\}$ for the full approach. For the lightweight approach, we adopt the learning rate and weight decay settings from DISEF(da Costa et al., 2023). We run the K-means clustering step for 300 iterations using the FAISS library (Douze et al., 2024) in the full approach. For the classifier tuning phase, we train for 50 epochs for the full approach and 150 epochs for the lightweight approach. The $\lambda_1, \lambda_2$ values are fixed to be $(0.1, 1)$ for the lightweight version and search inside $\{(0.1, 1), (2, 20), (20, 200), (50, 500)\}$ for the full version. In general, we find that our method is quite robust with this choice of values, no substantial performance difference observed when changing these values inside the defined set.

## C. Lightweight version

---

**Algorithm 2** Lightweight version

---

**Input**: Real dataset $\boldsymbol{S}$, number $g$ of synthesis samples, (conditional) Pretrained generator models $\mathcal{G}$, Learning rate schedule $\eta$

**Output**: Set of $g$ generated data samples $\boldsymbol{G}$

  1: Initialize centroids $\boldsymbol{z}$ for every local area
  2: Initialize noise vectors $(\boldsymbol{u}_1, \boldsymbol{u}_2, \dots, \boldsymbol{u}_g)$ randomly
  3: **for each** iteration **do**                                                        ▷ In each epoch
  4:     **for** mini-batch $A$ **do**
  5:         Sample real data set $\boldsymbol{S}_A$, and take their labels as conditional inputs for generator
  6:         **if** iteration = 1 **then**                                          ▷ Start optimize partition
  7:             Generate set $\boldsymbol{G}_A = \mathcal{G}(\boldsymbol{u}_A)$ based on labels condition of $\boldsymbol{S}_A$
  8:         **else**
  9:             Take set $\boldsymbol{G}_A$ from stored generated set.
10:         **end if**
11:         Assign data points $\boldsymbol{G}_A$ to their nearest clusters, indexed by $i$ and centered at $\boldsymbol{z}_i$
12:         Update learning rate $\eta \leftarrow \frac{1}{|\boldsymbol{z}_i|}$
13:         Update the center: $\boldsymbol{z}_i \leftarrow (1 - \eta)\boldsymbol{z}_i + \eta\boldsymbol{G}_k$
14:         Update $\boldsymbol{T}_S$ and the counts $g_i$
15:         Use own generated data to compute second term of loss function
16:         Compute loss function $\mathcal{L}$
17:         Train model $\boldsymbol{h}$ with loss function $\mathcal{L}$ and set of real data $\boldsymbol{S}_A$ and $\boldsymbol{G}_A$
18:     **end for**

---

The loss function differs slightly from the full version, where we compute regularization terms in all data instead of each batch. The reason for this is because the number of generated data is much smaller, so this computation ensures that the regularization is big enough to be meaningful. Initially, the algorithm optimizes with respect to data partitions (addressing the partition optimization problem) and subsequently refines classifiers . This is achieved using an alternate optimization strategy in each epoch to minimize the loss function $\mathcal{L}$ which was constructed as a combination of distribution matching, robustness terms, and classification loss. Note that in this version, the LoRA modules are also attached to the generator, without tuning them.

**Algorithm Overview**:

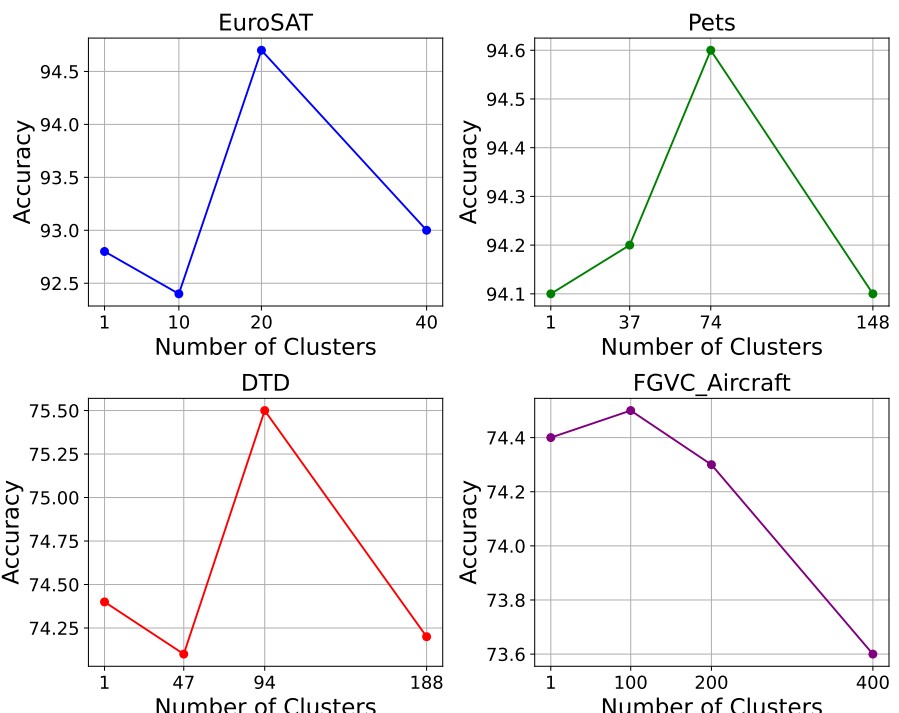

*Figure 3.* Results with increasing number of clusters on 4 datasets

1. **Initialization**:Begin with initializing the partitions and noise vectors essential for the iterations. Each iteration is one epoch of training classifier.

2. **Iterative Optimization**:

   - Partition Optimization: Utilize the MiniBatch K-means algorithm to optimize the partition (lines 11-13). Following partition updates, recalibrate dependent quantities (line 14) and monitor changes in regions to optimize memory usage when calculating $\mathcal{L}$.
   - Model Optimization: Forward real and synthetic data through models and update by loss function
   - Loss Computation and Gradient Descent: Calculate the loss function $\mathcal{L}$ based on computed components: output of real and synthetic data and their discrepancy. Finally, update model $h$ via gradient descent by backpropagating.

## D. Partitioning Experiments

In this section, we show the detailed figure of the analysis on number of clusters (Figure 3). The experiments were conducted on 4 datasets: EuroSAT, Oxford-IIIT Pets, DTD, and FGVC Aircraft, with the number clusters increased from 1 (all data belong to the same partition) to 1,2 and 4 times the number of classes in each dataset. The results validate our claim in the main text that the optimal choice of number of clusters typically about twice as the number of classes.

## E. More extreme few-shot scenarios

We conduct experiments on 3 datasets that were also used for DataDream (DD) (Kim et al., 2024). The results are shown in Table 4.

Overall, our method underperforms the baseline in the extreme 1-shot scenario. With only one real sample, the regularization terms in our loss function become small, reducing model robustness and possibly causing performance drops. However, our method significantly outperforms the baseline in 4-shot and 8-shot settings. Thus, extremely limited real data case remains a limitation of our approach.

| No. of real shots | AirC | | Cars | | FLO | |
|---|---|---|---|---|---|---|
| | DD | Ours | DD | Ours | DD | Ours |
| 1 | **31.1** | 25.3 | **72.9** | 72.1 | **88.7** | 86.1 |
| 4 | 38.3 | **51.6** | 82.6 | **86.9** | 96.0 | **96.9** |
| 8 | 54.6 | **63.9** | 87.4 | **91.3** | 98.4 | **98.7** |

*Table 4.* Results for more extreme few-shot conditions

## F. Adaptation to synthetic data only

In this section, we investigate a possible adaptation of our method to the case of synthetic data only. In order to do it, one can remove the discrepancy term and loss on real data from the loss function and compute the robustness loss on all regions that contains at least 2 synthetic samples. Overall, the loss function can be rewritten as follows: $F(\mathbf{G}, \boldsymbol{h}) + \lambda_2 \frac{1}{g} \sum_i \sum_{\boldsymbol{g_1},\boldsymbol{g_2} \in \mathbf{G_i}} \frac{1}{g_i} \|\boldsymbol{h}((\boldsymbol{g_1}) - \boldsymbol{h}(\boldsymbol{g_2})\|$. We conduct experiments to test the effectiveness of this loss function in some small and medium-sized datasets. The results are shown in Table 5.

| Dataset | DD | Ours |
|---|---|---|
| EuSAT | 80.3 | **80.6** |
| Pets | **94.0** | **94.0** |
| AirC | **71.2** | 70.6 |
| CAL | 96.2 | **96.8** |
| Food | 86.7 | **89.2** |

*Table 5.* Results if only synthetic data were used.

Our method outperforms the baseline on 3 out of 5 datasets, comparable in 1 and worse in 1 dataset. On average, our methods still perform better than the baseline, showcasing the necessity of the robustness regularization. However, these increases are marginal, and much less significant compared to our full method, which takes into account both discrepancy and robustness terms.

## G. Varying the number of synthetic data

To further investigate the effect of the number of synthetic samples, we conduct more experiments varying the number of them in Table 6. The results confirmed that our method consistently outperform baselines when varying synthetic data sizes.

| No. synth. samples | EuSAT | | DTD | | AirC | |
|---|---|---|---|---|---|---|
| | DD | Ours | DD | Ours | DD | Ours |
| 100 | 93.4 | **94.2** | 73.4 | **73.9** | 68.5 | **69.6** |
| 200 | 93.5 | **94.5** | 73.1 | **74.0** | 69.3 | **71.9** |
| 300 | 93.7 | **94.4** | 73.5 | **73.8** | 70.9 | **73.0** |
| 400 | 93.8 | **94.4** | 74.1 | **74.2** | 70.8 | **73.3** |
| 500 | 93.5 | **94.7** | 74.1 | **74.5** | 72.3 | **74.3** |

*Table 6.* The impact of the number of synthetic samples per class. Results of only the 16-shot real data

