# OpenReview forum: "Provably Improving Generalization of Few-shot models with Synthetic Data"
_ICML.cc/2025/Conference — ICML 2025 poster_

### Official Review · Reviewer_dDLg · 2025-03-11

**Overall Recommendation:** 3

**Summary:**

The paper presents a theoretical framework and corresponding algorithm to enhance the generalization of few-shot learning models by leveraging synthetic data.  Guided by our theoretical generalization error bounds, the authors introduce a novel loss function and training paradigm designed to jointly optimize data partitioning and model training. Experiment results demonstrate superior performance on the few-shot classification task.

**Claims And Evidence:**

The main claims regarding the benefits of minimizing distributional discrepancy and enhancing local robustness to improve generalization are convincingly supported through theoretical analysis and empirical evaluations.

**Essential References Not Discussed:**

Although the paper compares several related methods, many recent studies are still not discussed or compared, such as [r1], [r2], and [r3]. Thus, the related work section remains relatively weak.  [r2] also generates synthesis images.

[r1] Large Language Models are Good Prompt Learners for Low-Shot Image Classification. CVPR 2024.

[r2] Prompt, Generate, then Cache: Cascade of Foundation Models makes Strong Few-shot Learners . CVPR 2023.

[r3] Self-regulating Prompts: Foundational Model Adaptation without Forgetting. ICCV 2023.

**Experimental Designs Or Analyses:**

The experimental design is reasonable, using public datasets.  However, I have some questions in the **Weaknesses**.

**Methods And Evaluation Criteria:**

The proposed ProtoAug method is suitable for few-shot learning and the experimental setup is fair.

**Other Comments Or Suggestions:**

The sentence "From the theoretical perspective, existing studies typically have preliminary answers to simple models only: E.g., Yuan et al. (2024) proposed a framework that generates synthetic samples by training/finetuning a generator to minimize the distribution gap, addressing the second question." sounds awkward, particularly the use of "E.g." Consider revising the sentence structure.

**Other Strengths And Weaknesses:**

**Strengths：**
1. Synthetic data benefits few-shot learning by improving generalization.
2. The theoretical proofs provided are clear.

**Weaknesses：**
1. The experiments are insufficient, including the selection of comparative methods and additional experiments that require further validation. See the "Questions" section.
2. The authors do not provide a detailed hyperparameter analysis, such as the selection of weighting factors for the loss terms. How these hyperparameters were set is not clearly described, and the supplementary material does not seem to clarify this sufficiently.
3. The layouts of Figures 1 and 2 are not visually appealing.
4. The prior knowledge about the choice of Stable Diffusion (SD) is missing. Why was it selected specifically?

**Questions For Authors:**

1. Why didn't the authors report results for more extreme few-shot conditions (e.g., 1-shot, 4-shot, 8-shot)? These settings are commonly evaluated in few-shot learning literature.
2. What would the results be if only synthetic data were used?
3. How was the number of synthetic samples per class set to 500? Would using more or fewer samples have an impact? This seems to have a significant effect. Additionally, what would be the results if only the 16-shot real data were used?
4. Regarding the generation of synthetic data for the FGVC Aircraft dataset, is there a better solution?

**Relation To Broader Scientific Literature:**

Synthetic data and prototype learning have advanced the broader development of computer vision, providing valuable insights.

**Theoretical Claims:**

I thoroughly examined the theoretical results, especially Theorems 3.3 and 3.4.

---

> ### Author Rebuttal · Authors · 2025-03-31
>
> We would like to thank the reviewer for your positive evaluation and valuable feedback to our work. We would like to address your remaining concerns as follows:
>
> *1. Results for more extreme few-shot conditions*
>
> Thanks for your insightful suggestion. We conduct experiments on 3 datasets that was also used in the DataDream (DD) paper. The results are shown in the following table:
> |No. of real shots|AirC||Cars||FLO||
> |-|-|-|-|-|-|-|
> ||DD|Ours|DD|Ours|DD|Ours|
> |1|**31.1**|25.3|**72.9**|72.1|**88.7**|86.1|
> |4|38.3|**51.6**|82.6|**86.9**|96.0|**96.9**|
> |8|54.6|**63.9**|87.4|**91.3**|98.4|**98.7**|
>
> Overall, our method underperforms the baseline in the extreme 1-shot scenario. With only one real sample, the regularization terms in our loss function become small, reducing model robustness and possibly causing performance drops. However, our method significantly outperforms the baseline in 4-shot and 8-shot settings. Thus, extremely limited real data case remains a limitation of our approach.
>
> *2. Results if only synthetic data were used*
>
> We want to gently point out that our method are designed to take into account both real and synthetic data. Our loss function was designed to match between prediction of real and synthetic samples and regularization are computed on regions containing real data only. In order to adapt our method into the case of only synthetic data, one can remove the discrepancy term and loss on real data from the loss function and compute the robustness loss on all regions that contains at least 2 synthetic samples. Overall, the loss function can be rewritten as: $F(\mathbf{G},\boldsymbol{h})+\lambda_2\frac{1}{g}\sum_{i}\sum_{\boldsymbol{g_1},\boldsymbol{g_2}\in \mathbf{G_i}}\frac{1}{g_i}\\|\boldsymbol{h}((\boldsymbol{g_1})-\boldsymbol{h}(\boldsymbol{g_2})\\|$.
>
> We conduct experiments to test the effectiveness of this loss function in some small and medium-sized datasets. The results are shown below:
>
> |Dataset|DD|Ours
> |-|-|-
> |EuSAT|80.3|**80.6**
> |Pets|**94.0**|**94.0**
> |AirC|**71.2**|70.6
> |CAL|96.2|**96.8**
> |Food|86.7|**89.2**
>
> Our method outperforms the baseline on 3 out of 5 datasets, comparable in 1 and worse in 1 dataset. On average, our methods still perform better than the baseline, showcasing the necessity of the robustness regularization. However, we admit that these increases are marginal, and much less significant compared to our full method, which takes into account both discrepancy and robustness terms.
>
> *3. The impact of the number of synthetic samples per class. Results of only the 16-shot real data*
>
> We chose 500 to be consistent with the main experiment in the DataDream paper, ensuring a fair comparison in Tables 1 and 3.
>
> To further investigate the effect of the number of synthetic samples, we conduct more experiments summarized below:
>
> |No. synth. samples|EuSAT||DTD||AirC||
> |-|-|-|-|-|-|-|
> ||DD|Ours|DD|Ours|DD|Ours|
> |100|93.4|**94.2**|73.4|**73.9**|68.5|**69.6**|
> |200|93.5|**94.5**|73.1|**74.0**|69.3|**71.9**|
> |300|93.7|**94.4**|73.5|**73.8**|70.9|**73.0**|
> |400|93.8|**94.4**|74.1|**74.2**|70.8|**73.3**|
> |500|93.5|**94.7**|74.1|**74.5**|72.3|**74.3**|
>
> The results confirmed that our method consistently ourperform baselines when varying synthetic data sizes.
> The results of only 16-shot real data was reported in the "Real-finetune" row in Table 1 in our paper.
>
> *4. Better solution for generation of synthetic data for the FGVC Aircraft datasets*
>
> Stable Diffusion is infamously bad on aircraft data. In practice, we can choose a better generative model on aircraft data. A future direction for our method involves using the theoretical framework for optimizing synthetic data by fine-tuning generator or filtering synthetic data by using our loss function as a criterion.
>
> *5. Detailed hyperparameter analysis*
>
> We want to gently redirect the reviewer's attention to our response to Reviewer hvE3 (part 4) to see our detailed discussion about the hyperparameter selection. [https://openreview.net/forum?id=L6U7nYc4ah&noteId=sANtK8HtxS]
>
> *6. Essential References Not Discussed*
>
> We thank the reviewer for highlighting missing references. We will include the following discussion of the mentioned papers and add further references on prompt and adapter tuning in the revised version:
>
> Recently, methods such as prompt and adapter tuning have become prominent for enhancing few-shot classification performance, including [r1], [r3], etc. Moreover, [r2] uses synthetic images to improve performance but relies on combining four large pretrained models, significantly increasing implementation complexity.
>
> *7. Not visually appealing figures and awkward sentence*
>
> Thanks for suggesting these issues, we will improve the presentation in the new version.
>
> *8. Stable Diffusion selection*
>
> We chose it to be consistent with the baseline for fair comparison. DataDream also uses Stable Diffusion of the same version as the generator.

---

### Official Review · Reviewer_L3yv · 2025-03-13

**Overall Recommendation:** 3

**Summary:**

The paper tackles the problem of few-shot image classification, where limited labeled data restricts model generalization, by proposing a theoretically grounded approach that augments real data with synthetic data, such as that produced by Stable Diffusion. It introduces a novel test error bound for models trained on mixed real and synthetic datasets, highlighting the need to minimize the distribution gap between data types and ensure local robustness around training samples. The authors develop a new loss function and training method that incorporates prototype learning via K-means clustering for data partitioning and regularization to align predictions and boost robustness. Experiments across 10 diverse datasets demonstrate that this method surpasses state-of-the-art baselines like DataDream, DISEF, and IsSynth, achieving an average performance boost of 0.6% and exceeding 2% on challenging datasets like Food101 and FGVC Aircraft, with ablation studies validating the effectiveness of the proposed components.

**Claims And Evidence:**

Supported Claim1 : The method outperforms state-of-the-art baselines across multiple datasets

Supported Claim2 : The lightweight version of the algorithm provides competitive performance with reduced computational costs

Supported Claim3 : The proposed algorithm, guided by the theoretical bound, effectively minimizes generalization errors by optimizing data partitioning and model training

**Essential References Not Discussed:**

N/A

**Experimental Designs Or Analyses:**

The experimental designs and analyses in the paper are largely sound and valid, supported by diverse datasets, strong baselines, and thorough evaluations.

**Methods And Evaluation Criteria:**

The proposed methods and evaluation criteria make sense for the problem of few-shot image classification with synthetic data. The theoretical framework tackles the critical issues of distribution gaps and data scarcity, and the algorithm builds on this with practical clustering and loss design. The implementation offers flexible options for performance and efficiency. Meanwhile, the evaluation uses diverse datasets, strong baselines, a suitable metric, and thorough ablation studies to demonstrate the method’s effectiveness. While minor enhancements—like more data on computational savings—could strengthen the case, the overall approach is well-tailored to the problem at hand.

**Other Comments Or Suggestions:**

N/A

**Other Strengths And Weaknesses:**

N/A

**Questions For Authors:**

N/A

**Relation To Broader Scientific Literature:**

N/A

**Theoretical Claims:**

Did not check

---

> ### Author Rebuttal · Authors · 2025-03-31
>
> We would like to thank the reviewer for praising our work and your positive evaluation. We would like to reiterate the key novelties of our work:
> 1. We introduced a novel generalization bound for a model trained with synthetic data augmentation. It suggests an effective way to generate synthetic samples, and to train a model by minimizing the distribution gap between real and synthetic data and ensuring local robustness around real samples.
> 2. Guided by this theory, we designed an algorithm to jointly optimize models and data partitioning to enhace models' performance.
> 3. Experimental results show that our method outperforms state-of-the-art baselines across multiple datasets. Our more efficient lightweight version also shows comparable performance with the baselines.
>
> Should you have any more questions or concerns, we would be very happy to address them.

---

### Official Review · Reviewer_hvE3 · 2025-03-16

**Overall Recommendation:** 3

**Summary:**

The paper addresses the challenge of improving few-shot image classification by augmenting real data with synthetic data. It identifies the distribution gap between real and synthetic data as a key obstacle. The paper presents a theoretical framework to quantify the impact of this distribution discrepancy on supervised learning and proposes a theoretically-driven algorithm that integrates prototype learning to optimize data partitioning and model training. The algorithm aims to bridge the gap between real and synthetic data. The authors claim that their method outperforms state-of-the-art approaches across multiple datasets. The key idea is to minimize a generalization error bound that accounts for both the discrepancy between the real and synthetic distributions and the robustness of the predictor around training samples. They achieve this through a novel loss function and a training paradigm that jointly optimizes data partitioning and model training.

**Claims And Evidence:**

The overall claim that the proposed method improves few-shot learning with synthetic data is supported by the experimental results on several datasets. The tables presented show superior performance compared to several baselines, including DataDream, DISEF, and IsSynth.

The individual contributions of different components (data partitioning, discrepancy minimization, robustness) are somewhat validated through ablation studies. However, the ablation study in Table 2 only shows the effect of regularization on a few datasets. A more comprehensive ablation study on more datasets might be better to support the effectiveness of each component.

**Essential References Not Discussed:**

No

**Experimental Designs Or Analyses:**

The experimental designs seem generally sound.

Ablation Studies: The ablation studies in Table 2 are helpful for understanding the impact of different components of the algorithm. However, the ablation study only focus on three datasets.

Hyperparameter Sensitivity: The paper mentions tuning hyperparameters. A more detailed discussion of the hyperparameter selection process would be useful. It would be beneficial to see a sensitivity analysis that shows how the performance of the algorithm varies with different hyperparameter values. A plot of each hyperparameter in relation to the accuracy/generalization bound should be shown.

**Methods And Evaluation Criteria:**

The proposed method combines theoretical analysis with a practical algorithm. The K-means clustering for partitioning and the prototype learning aspect appear reasonable for the problem. The evaluation uses standard datasets for few-shot image classification (e.g., FGVC Aircraft, Caltech101, Food101), which is appropriate. Comparing against DataDream, DISEF, and IsSynth seems adequate given the focus on synthetic data augmentation.

**Other Comments Or Suggestions:**

The paper could benefit from a clearer statement of the assumptions made in the theoretical analysis.

A more detailed discussion of the hyperparameter selection process would be helpful.

**Other Strengths And Weaknesses:**

Strengths:

-The combination of theoretical analysis and a practical algorithm is a strength.

-The experimental results demonstrate promising performance.

Weaknesses:

-The tightness and practical implications of the theoretical bounds need further clarity.

**Questions For Authors:**

What are the key assumptions made in deriving the generalization bounds in Theorems 3.3 and 3.4?

Why do you choose the parameters in DataDream, DISEF and IsSynth as your baseline? Have you tuned the hyper-parameters to have a fair comparison?

In ablation studies, why not include results on the four datasets used in Table 3?

Why does lightweight version give you better performance for some datasets?

**Relation To Broader Scientific Literature:**

The paper relates to several areas of research, namely, Few-Shot Learning, Synthetic Data Augmentation, Domain Adaptation and Dataset Distillation

The paper makes explicit connections to DataDream, DISEF, and IsSynth. It would be useful to discuss how the proposed theoretical framework relates to existing theoretical work on domain adaptation or generalization with synthetic data.

**Theoretical Claims:**

While the authors derive generalization error bounds, the practical implications of these bounds depend on the tightness of the bounds and the validity of the assumptions made in their derivation.

What are the key assumptions made in deriving the generalization bounds (e.g., concerning the loss function, the data distributions, the model class)? Are these assumptions realistic for the problem of few-shot image classification with synthetic data?

Generalization bounds are often loose. How tight are the derived bounds in practice? Is there any discussion of the practical implications of the bounds (e.g., how the different terms in the bound influence the performance of the algorithm)?

---

> ### Author Rebuttal · Authors · 2025-03-31
>
> We thank the reviewer for your positive evaluation and valuable suggestions. We address your concerns below.
>
> *1. Key assumptions made in deriving the generalization bounds, and their realisticity*
>
> The only assumptions are (1) data samples are i.i.d. and (2) loss function is bounded and Lipschitz w.r.t. the model. Note that for classification problems, many common losses satisfy our assumptions, such as absolute loss, square loss, Hinge loss. Therefore, those assumptions are realistic and reasonable, widely used in the literature.
>
> *2. Tightness and practical implications of the bounds*
>
> *Tightness:* We consider several factors regarding the tightness of our bounds:
>
> - Our bound scales with $O(\sqrt{K})$, which can be sub-optimal for large $K$ (partition size). Therefore, we should not choose a large $K$. Luckily, for few-shot problems, our bound does not allow a large $K$ which goes beyond the number of real samples.
> - For fixed $K$ and number $n$ of real samples, our bound scales with $O(g^{-0.5})+const$. This indicates *our bound is optimal w.r.t. the number of synthetic samples.* The reason is that the error of the best model is at least $O(g^{-0.5})+const$, for a hard learning problem, according to Theorem 8 in [Than et al. (2024). Gentle local robustness implies Generalization].
> - Finally, since $n$ can be very small, our bound can be far from the true error of a model. This is a common limitation of any theoretical error bound for limited data. To the best of our knowledge, *limited data poses a big (open) challenge* for estimating the model's true error.
>
> *Practical implications:* We have discussed various implications after Theorem 3.3 in the paper. We point out some useful roles of the main terms. We also have Sections 5.3 and 5.4 to investigate empirically the influence of robustness and discrepancy terms.
>
> *3. Ablation study limited to 3 datasets*
>
> We conducted ablation study on 4 datasets of varying sizes,  which we believe sufficiently illustrate our method's effectiveness. We will add more results on the remaining datasets in the Appendix.
>
> *4. Hyperparameter selection process*
>
> First, we balance the discrepancy ($\lambda_1$) and the robustness ($\lambda_2$), using coordinate descent to determine this optimal ratio. We then chose a suitable value of $\lambda$ associated with the loss on real data. The results are reported below:
> |||DTD||EuSAT||Pets||
> |-|-|-|-|-|-|-|-
> |||light|full|light|full|light|full
> |**dis/rob ratio** |1/1|69.2|72.6|89.9|**93.9**|93.1|94.3
> ||1/5|72.8|72.8|93.0|93.8|**94.5**|94.3
> ||1/10|**73.7**|**73.1**|**93.3**|**93.9**|**94.5**|94.3
> ||1/20|73.2|72.8|92.6|**93.9**|94.3|**94.4**
> |$\lambda$|1|73.7|73.1|93.3|93.9|94.5|94.3
> ||2|74.9|73.4|93.3|94.1|94.3|94.5
> ||4|**75.2**|**74.5**|**93.8**|**94.7**|**94.8**|94.6
> ||6|75.1|74.4|92.7|94.3|94.5|**94.8**
>
> We then selected the actual values of ($\lambda_1$/$\lambda_2$) from a set of values that follows the 1/10 ratio. We chose their values to maintain a good ratio between the regularization and loss.
>
> For lightweight version, we directly adopted the learning rate and weight decay from the DISEF paper, as our synthetic sample sizes matched theirs. For the full version, we employed grid search with early stopping to choose them.
>
> These hyperparameter value sets are presented in Appendix B. The others follow exactly as in the baselines.
>
> Note that hyperparameter selection was performed only during classifiers tuning, separately from generator tuning and synthetic data generation, minimizing additional computational overhead.
>
> *5. Relation to existing theoretical work on domain adaptation or generalization with synthetic data*
>
> We thank the reviewer for useful suggestion. We have discussed the existing theoretical works of generalization with synthetic data in the last paragraph of section 2.1. We will add more related references.
>
> *6. Why select parameters from DataDream (DD), DISEF, and IsSynth as baselines? Were hyperparameters tuned for fair comparison?*
>
> We chose hyperparameters to be consistent with DD. Since DD's authors reran DISEF and IsSynth on the same number of synthetic data and generative models, it serves as a good baseline for us. Additionally, experiments confirmed that directly using baselines' learning rate and weight decay yielded optimal results across datasets.
>
> *7. In ablation studies, why not include results on the 4 datasets used in Table 3?*
>
> We followed the Resnet50 setup from DD in table 3 for fair comparison. However, in the ablation study, we aim to evaluate our method's performance across datasets of varying scales, whereas the datasets in Table 3 are similar in size.
>
> *8. Lightweight version give better performance for some datasets*
>
> The lightweight version performs better on 2 datasets: Caltech and DTD. For these two datasets, we observed that the methods with generator fine-tuning (DD and our full version) perform equal or worse than the others without it (DISEF, IsSynth, and our lightweight).

---

### Decision · Program_Chairs · 2025-05-01

**Decision:**

Accept (poster)

**Comment:**

The final ratings are 3 weak accept.

Reviewer hvE3 is mainly concerned about the practical implications of the error bound. The authors have addressed those issues in the rebuttal. Reviewer L3yv generally appreciates the idea of this work. Reviewer dDLg, though gave a positive rating, requested more experimental comparison on extreme settings. The authors provided such experiments in the rebuttal and concluded that the proposed method is not suitable for the extreme 1-shot scenario.

After checking the review, the manuscript, and the rebuttal, the AC notices that the performance gain brought by the proposed method is not that significant, especially considering the experiments requested by the reviewers. However, the AC believes that the authors did a good rebuttal and the work is worth sharing with a broader audience.